# Towards Rationale-Answer Alignment of LVLMs via Self-Rationale Calibration

**Yuanchen Wu** [1 2 *]   **Ke Yan** [2]   **Shouhong Ding** [2]   **Ziyin Zhou** [3 2]   **Xiaoqiang Li** [1]

## Abstract

Large Vision-Language Models (LVLMs) have manifested strong visual question answering capability. However, they still struggle with aligning the rationale and the generated answer, leading to inconsistent reasoning and incorrect responses. To this end, this paper introduces **Self-Rationale Calibration (SRC)** framework to iteratively calibrate the alignment between rationales and answers. SRC begins by employing a lightweight "rationale fine-tuning" approach, which modifies the model's response format to require a rationale before deriving answer without explicit prompts. Next, SRC searches a diverse set of candidate responses from the fine-tuned LVLMs for each sample, followed by a proposed pairwise scoring strategy using a tailored scoring model, R-Scorer, to evaluate both rationale quality and factual consistency of candidates. Based on a confidence-weighted preference curation process, SRC decouples the alignment calibration into a preference fine-tuning manner, leading to significant improvements of LVLMs in perception, reasoning, and generalization across multiple benchmarks. Our results emphasize the rationale-oriented alignment in exploring the potential of LVLMs.

## 1. Introduction

Recently, with the advancement of large language models (LLMs) (Dubey et al., 2024; Yang et al., 2024), the integration of visual encoders through multimodal alignment pre-training and instruction fine-tuning has enabled Large Vision-Language Models (LVLMs) to achieve significant task-level generalization (Liu et al., 2024a; Wang et al.,

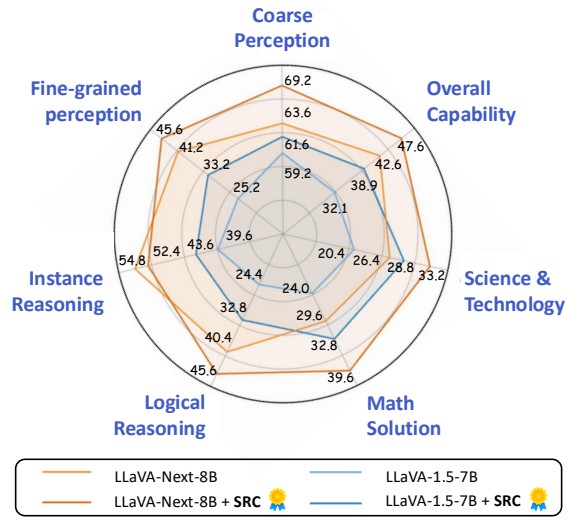

*Figure 1.* **SRC on different LVLMs.** The evaluation dataset is MMStar. The baselines are LLaVA-1.5-7B and LLaVA-Next-8B.

2024b). These advancements have expanded various applications, such as image captioning, visual chat, and visual question answering (VQA). However, the misalignment of LVLMs between their vision and text modalities (Cui et al., 2023; Zhu et al., 2024), *i.e.*, the model outputs some text descriptions that do not match visual elements, is a critical issue for practical applications. Many studies have been exploring **post-training** strategies, leveraging additional supervised fine-tuning (SFT) (Liu et al., 2023; Zhang et al., 2025) or conducting preference alignment (Zhu et al., 2024; Zhou et al., 2024b) to overcome this issue. The latter, particularly through Direct Preference Optimization (DPO) (Rafailov et al., 2024) to **discourage** models from generating counterfactual descriptions of images, has emerged as a popular paradigm. Many works collect preference data in diverse ways, such as perturbing image descriptions (Zhou et al., 2024a), introducing expert models (Zhao et al., 2023), or sampling from outputs (Zhou et al., 2024b), and achieve promising results. However, this form of alignment, while effective for visual description, **overlooks the rationales essential for generating factually grounded and logically consistent responses**, especially in VQA scenarios.

Moreover, the current instruction fine-tuning process (Liu et al., 2024c) predominantly relies on datasets composed of

*Work done during internship at Tencent Youtu Lab. [1]School of Computer Engineering and Science, Shanghai Univeristy [2]Tencent Youtu Lab [3]Key Laboratory of Multimedia Trusted Perception and Efficient Computing, Xiamen University. Correspondence to: Ke Yan <kerwinyan@tencent.com>, Xiaoqiang Li <xqli@shu.edu.cn>.

*Proceedings of the 42nd International Conference on Machine Learning*, Vancouver, Canada. PMLR 267, 2025. Copyright 2025 by the author(s).

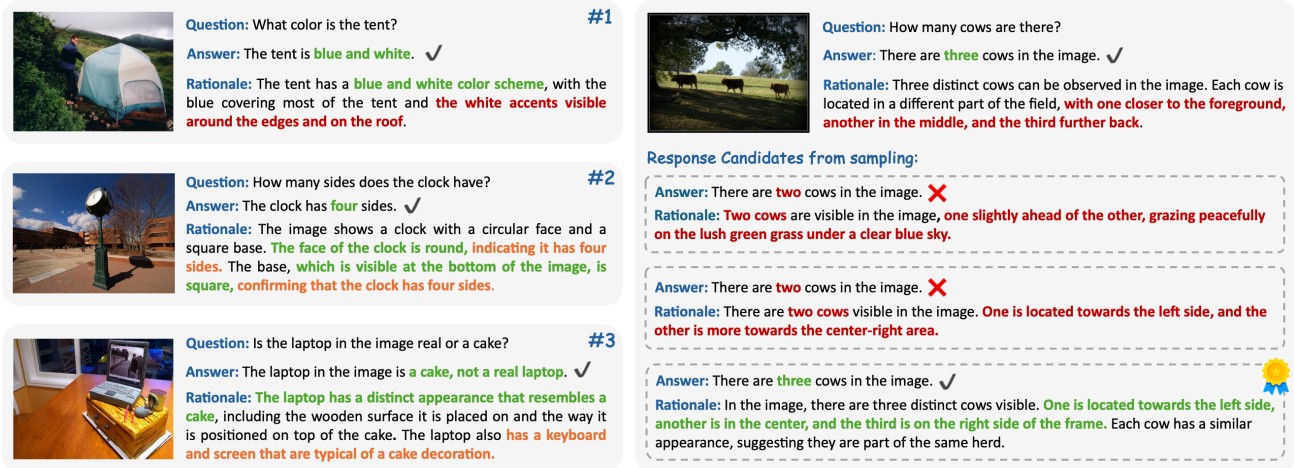

*Figure 2.* **Examples of model responses in VQA scenarios.** Though the answers and visual elements are often correct (green), the rationales may be counterfactual (red) or insufficient (orange). Rationales are generated using LLaVA-1.5-7B in multi-round dialogues.

short answers. This could result in models lacking explicit rationale supervision, potentially leading to spurious causal associations between instructions and responses, rather than fostering a vision-driven thinking process in models. Hence, in this paper, we seek to further explore **the alignment between a** *model's rationale and its answers*, and consider a critical question: "*Does a correct answer stem from a reasonable and factually grounded rationale?*"

To dive into this question, we prompt LVLMs to provide rationales alongside their answers and assess their alignment quality. As illustrated in Figure 2, although models often produce correct answers, their rationales may be **counterfactual (#1)**, **insufficient (#2)**, or **unreasonable (#3)** to support the answers. Also, we can observe a **notable uncertainty** of the sampled responses in terms of the answers with rationales. To this end, we propose a *"rationale-oriented"* preference alignment framework, named "**Self-Rationale Calibration (SRC)**". The **key insight** of SRC is that for any given vision instruction (*e.g.*, a VQA sample), the model's output space inherently contains a spectrum of rationale-answer pairs (RAPs), ranging from *relatively optimal pairs* (*e.g.*, a rationale that is factually accurate and logically consistent with the answer) to *relatively inferior pairs* (*e.g.*, a rationale that is counterfactual or invalid for the answer), regardless of whether final answers are consistent with ground-truth. By leveraging the model's inherent output space, SRC distinguishes between optimal and inferior RAPs to iteratively calibrate the model itself, progressively aligning the model's rationales with its answers.

Specifically, we first augment partial VQA samples to construct RAPs and then fine-tune LVLMs using a lightweight LoRA (Hu et al., 2021), inducing the model to **provide RAPs without explicit prompting**. Using this variant as the *seed* model, we perform sentence-level beam search in

the output space to generate diverse response candidates for each visual instruction sample. Then, we introduce **a pairwise scoring strategy** with **a tailored LLM-based scoring model** named "**R-Scorer**" to evaluate these various candidates. The motivation behind this strategy lies in the open-ended nature of rationales: even among candidates with correct answers, their rationales may vary in quality. By performing pairwise scoring coupled with "LLM-as-judge" (Zheng et al., 2023), SRC can effectively capture the **"relative superiority"** between rationales among candidates, such as the reasoning process and logical relationships between visual elements in the images. To further address challenges such as neutral scoring results, which may obscure differences between candidates, and to mitigate potential scoring biases, we **aggregate the confidence scores** of the candidates to identify both optimal and inferior candidates. Finally, the model's rationale generation process is calibrated through preference alignment, which helps to improve its vision-driven thinking process. By iteratively applying the above process, the model progressively improves its perception and reasoning abilities, leading to enhanced overall performance.

Overall, our contributions are summarized as follows:

● **Rationale-oriented Post-training Framework.** We propose a novel framework, Self-Rationale Calibration (SRC), to address the misalignment between rationales and answers in LVLMs. SRC iteratively enhances models by leveraging both optimal and inferior RAP candidates, focusing on the relation between rationale and answer correctness.

● **Pairwise Scoring Strategy with R-Scorer.** We develop a pairwise scoring strategy with a lightweight scoring LLM named R-Scorer to assess the quality of response candidates. This allows for relative comparisons between candidates,

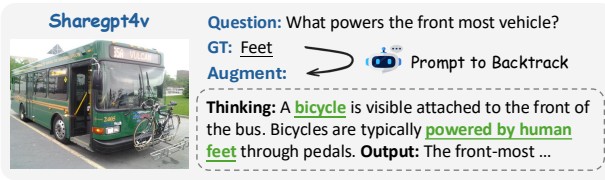

Figure 3. **One rationale-augmented sample in our dataset.** The augmented sample has sufficient rationale to support GT.

1. ☑ **Validity:** the rationale should support the answer;

2. ⊞ **Coherence:** the rationale should exhibit logical consistency and structural clarity;

3. 🔍 **Rationality:** the rationale should conform to the world knowledge, such as math and science;

Figure 4. **The criteria of data construction** in Rationale Fine-tuning: *validity*, *coherence*, and *rationality*.

enabling robust identification of optimal and inferior candidates for preference fine-tuning.

• **Enhanced Comprehensive capabilities of LVLMs.** Through extensive experiments, we demonstrate the effectiveness of SRC. Compared to post-training techniques of vision-text alignment, SRC significantly improves performance in QA scenarios (as shown in Figure 1), achieving strong generalization and reasoning capabilities.

## 2. Related Work

**Preference and Modality Alignment of LVLMs.** Preference alignment is a post-training paradigm widely adopted in recent LLMs, including methods such as PPO (Schulman et al., 2017) and DPO (Rafailov et al., 2024), to align models with human preferences. In LVLMs, the inherent misalignment between vision and text modalities—where textual descriptions fail to correspond to visual elements—has led many recent approaches to leverage the popular DPO method to alleviate this issue. Some studies rely on human annotators (Li et al., 2023b; Yu et al., 2024a) or expert models (Zhao et al., 2023; Zhou et al., 2024b) to curate preferred samples, while others introduce perturbed images to generate non-preferred samples (Deng et al., 2024; Zhou et al., 2024a). Overall, these methods focus on the *image captioning task* when constructing preference samples to promote vision-text alignment through DPO. Instead, SRC targets on the alignment between *rationales and answers*, especially in VQA scenarios. Specifically, rather than merely ensuring the correctness of visual element descriptions, SRC further emphasizes reasoning and logical relationships between visual elements and instructions (questions).

**Chain-of-Thought (CoT) Learning.** CoT enables step-by-step reasoning of LLMs for tackling complex questions (Wei et al., 2022). Since OpenAI-o1 (OpenAI, 2024b), some studies of LVLMs have been exploring autonomous and structured reasoning in vision-language tasks through distillation. They generate high-quality reasoning paths from expert models (*e.g.*, GPT-4o (Xu et al., 2024)) through explicit CoT prompting, where the prompt is later removed during SFT to allow models to mimic CoT reasoning autonomously (Zhang et al., 2024; Guo et al., 2024). While they share similarities with rationale fine-tuning, it is noted that they emphasize *long step-by-step CoT reasoning path*

by learning from large-scale datasets (from 100K to 12M samples). In contrast, rationale fine-tuning of SRC is a lightweight strategy that modifies models to explicitly output rationales during question answering. The fine-tuned variant will be used as the seed model for subsequent alignment of the rationale and answer through calibration.

## 3. Method: Self-Rationale Calibration

This section presents the proposed **Self-Rationale Calibration (SRC)** framework to enhance the alignment of the factual and reasonable rationales with corresponding answers. As depicted in Figure 5, it consists of *four* stages, *i.e.*, "Rationale Fine-tuning", "Pairwise Candidate Scoring", "Confidence-weighted Preference Curation", and "Calibration via Preference Fine-tuning". Finally, we conduct the above process to calibrate the model through multiple iterations, achieving both continual alignment and improvement of the model. To address the challenge of evaluating candidates and scoring efficiency in the Pairwise Scoring stage, we further developed a lightweight LLM-based scorer, named R-Scorer, tailored for SRC framework.

### 3.1. Rationale Fine-tuning

**Motivation.** In the instruction fine-tuning stage, LVLMs are heavily supervised with VQA samples with short answers (Hudson & Manning, 2019; Masry et al., 2022). The models tend to directly output brief answers, often requiring prompting to generate specific rationales. Thus, we introduce "rationale fine-tuning" to induce the model to provide a rationales before providing a answer (*i.e.*, RAPs) **without explicit prompting**. After that, we use the fine-tuned variant to promote the alignment of answers and rationales to improve the perceptual and reasoning performance.

**Data Construction.** We begin by collecting and sampling publicly available VQA datasets, and augmenting them with rationales, as shown in Figure 3. Specifically, we select `57k` samples to build the data pool (details can be viewed in Section 4.1), including three categories: ***perception & world knowledge***, ***chart understanding***, and ***math & science***. Then, we augment these samples using an advanced LVLM, `Qwen2-VL-72B` (Wang et al., 2024b), with a uni-

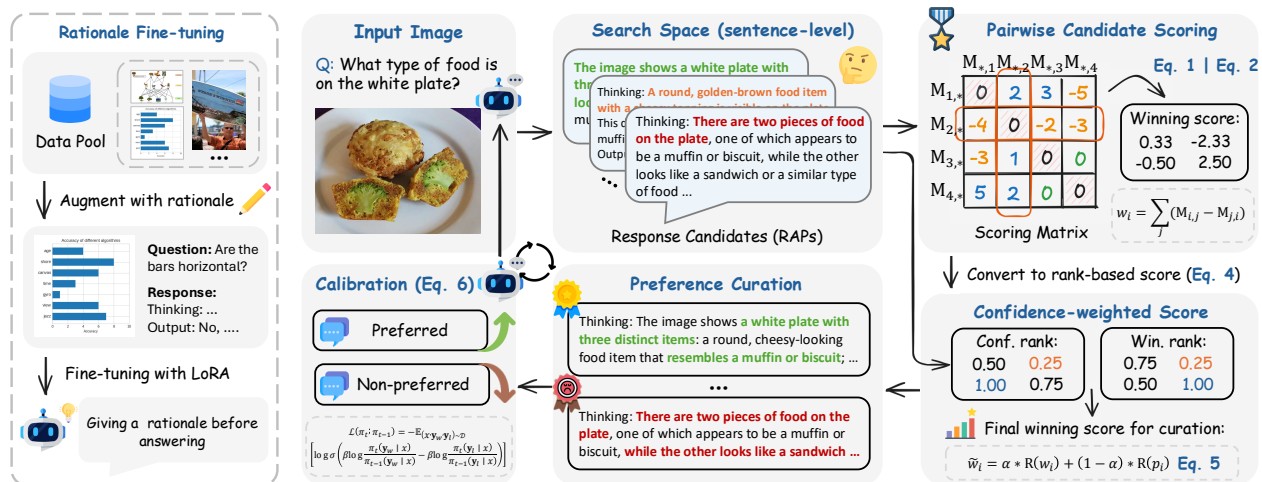

*Figure 5.* **Overview of SRC framework**. The framework consists of four iterative stages: "Rationale Fine-tuning", "Pairwise Candidate Scoring", "Confidence-weighted Preference Curation", and "Calibration via Preference Fine-tuning".

fied response format: "*Thinking: (rationale) + Output: (answer)*", serving as the dataset for fine-tuning, calibration, and evaluation in our work. In this process, Qwen2-VL is prompted with GT for each sample to backtrack the corresponding rationale. The intermediate rationales are subsequently filtered using Qwen2.5-72B (Yang et al., 2024) based on the three criteria in 4. This filtering process resulted in a final dataset of approximately 43K samples. The specific prompts employed in this data construction process are provided in Appendix A.1.

**Learning Rationales.** Ghosh et al. (2024) demonstrate that LoRA fine-tuning does not significantly alter the model's internal knowledge but instead adjusts the response format. Based on this insight, we adopt a lightweight LoRA fine-tuning approach (*e.g.*, rank=4 for adapting LLaVA-1.5 (Liu et al., 2024a)). We empirically set a 2:1:1 sampling ratio across the above three categories, and construct a set of nearly 20K samples for rationale fine-tuning. The fine-tuned models will be used as the ***seed models*** for the subsequent self-rationale calibration process.

### 3.2. Pairwise Candidate Scoring

**Motivation.** Though some prior methods focus on assessing answer correctness (Zelikman et al., 2022) or independently scoring (Wang et al., 2024d) to identify the optimal response and post-train LVLMs, they would face limitations when applied to RAPs. One major issue is that, even among candidates with correct answers, their rationales can vary significantly, making it difficult to determine their quality using GT. Moreover, relying on independent scoring fail to capture the ***"relative superiority"*** of rationales. To address these challenges, we propose a pairwise scoring approach coupled with "LLM-as-judge" (Zheng et al., 2023). Further discussions can be found in Appendix A.3.

**Candidate Generation.** By adopting beam search on sentence level (the details are provided in Appendix A.2), we generate multiple RAP response candidates in the data pool, each accompanied by ***rationales*** and their corresponding ***answers***. For each sample, we collect a candidate set denoted as $\mathbb{P}$. Considering the efficiency of pairwise scoring, we select the $N = \min(N, |\mathbb{P}|)$ most distinctive candidates based on their sentence embeddings (Chen et al., 2024b), where $|\cdot|$ represents the cardinality of a candidate set and $N = 6$ in our practical implementation.

**Pairwise Scoring.** As shown in Figure 6, the input of LLMs to conduct pairwise scoring includes three parts: ①"*Question with GT*", ②"*Candidate A with its factual checks*", and ③"*Candidate B with its factual checks*". The GT guarantees the correctness of the answer, while factual checks evaluate the vision-text alignment of rationales. For factual checks, LLMs pose three questions related to visual elements in the rationale and prompt seed models to engage in self-reflection checks, followed by He et al. (2024) and Cheng et al. (2024). The scoring prompt used for the LLM is provided in Appendix A.3. For each candidate pair $(\mathbf{C}_i, \mathbf{C}_j) \in \mathbb{P}$, its corresponding score $s_{i,j} \in [-5, 5]$ reflects the relative quality between the two candidates. A ***positive*** score indicates that $\mathbf{C}_i$ is ***better*** than $\mathbf{C}_j$, with a ***higher*** score implying a greater degree of superiority, while a ***negative*** score suggests the opposite. Our early experiments show that the order of candidates in the prompt significantly impacts the pairwise scoring results when using either proprietary or open-source LLMs. Thus, we apply ***bidirectional scoring*** by swapping the candidate order.

### 3.3. Confidence-weighted Preference Curation

**Calculation of Winning Score.** After performing $N(N-1)$ pairwise scoring processes, we construct a score matrix

$\mathbf{M} \in \mathbb{R}^{N \times N}$, where $\mathbf{M}_{i,j} = s_{i,j}$ for $i \neq j$ and $\mathbf{M}_{i,i} = 0$. Once the pairwise score matrix $\mathbf{M}$ is obtained, the **winning score** $w_i$ of each candidate $\mathbf{C}_i$ relative to all other candidates, is defined as:

$$w_i = \sum_j (\mathbf{M}_{i,j} - \mathbf{M}_{j,i}), \qquad (1)$$

where $j \in 1, \ldots, N$ and $j \neq i$. This score aggregates the difference in pairwise scoring for candidate $\mathbf{C}_i$. Additionally, we explore an alternative winning score based on the **count**, which is defined as:

$$w_i = \sum_j \mathbb{I}(\mathbf{M}_{i,j} > 0) + \sum_j \mathbb{I}(\mathbf{M}_{j,i} < 0), \qquad (2)$$

where $\mathbb{I}(\cdot)$ denotes the indicator function. $w_i$ reflects the **relative superiority** of $\mathbf{C}_i$ compared to others within its candidate set. However, multiple candidates have similar quality, which could lead LLMs to assign neutral scores during the scoring process. Also, the preference bias of LLMs may lead to unsatisfactory scoring process. To address these issues, we incorporate **candidate confidence** to calibrate $w_i$ introduced in the following section.

**Confidence Score of Candidates.** Each response candidate generated during sentence-level beam search is associated with a probability, which can be interpreted as the corresponding confidence score. It is derived from the language decoder of the model and represents the cumulative log-probability of generating a complete response. For candidate $\mathbf{C}_i$, its confidence $p_i$ is formulated as:

$$p_i = \prod_{t=1}^{\mathrm{len}(\mathbf{C}_i)} P(s_t \mid x, s_1, s_2, \ldots, s_{t-1}), \qquad (3)$$

where $\mathrm{len}(\mathbf{C}_i)$ denotes the number of sentences in $\mathbf{C}_i$, $x$ indicates the multi-modal input (*i.e.*, the input image and text prompt), and $s_t$ represents the $t$-th sentence within $\mathbf{C}_i$. A higher $p_i$ indicates the model's stronger preference for selecting the corresponding candidate as its response.

**Confidence-weighted Winning Score.** Since direct combination of $w_i$ with $p_i$ is challenging due to the inconsistent ranges and scales of these two scores, we first convert $w_i$ and $p_i$ to a unified **rank-based** score. Let $\{z_1, z_2, \ldots, z_N\}$ be a set of scores (*e.g.*, $\{w_1, \ldots, w_N\}$ or $\{p_1, \ldots, p_N\}$), we define a rank transformation function $\mathrm{R}(\cdot)$ that maps each score $z_i$ to a normalized rank in the interval $[0, 1]$. Specifically, we sort the scores in descending order and assign each score $z_i$ a rank, with rank 1 corresponding to the largest value. The transformation is defined as:

$$\mathrm{R}(z_i) = \frac{\mathrm{rank}(z_i)}{N}, \qquad (4)$$

where $\mathrm{rank}(z_i) \in \{1, 2, \ldots, N\}$ denotes the ordinal position of $z_i$ under descending order. After obtaining $\mathrm{R}(w_i)$

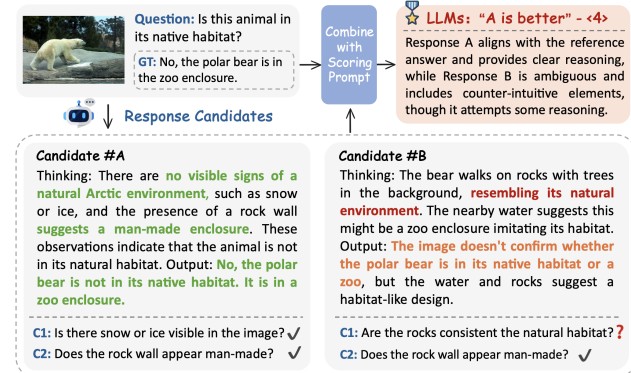

*Figure 6.* **The pairwise scoring process.** "C1/C2": the factual check question samples generated by the scoring LLMs.

and $\mathrm{R}(p_i)$ for each candidate $\mathbf{C}_i$, we combine them using a weighting factor $\alpha \in [0, 1]$, resulting in the **confidence-weighted** winning score $\widetilde{w}_i$:

$$\widetilde{w}_i = \alpha * \mathrm{R}(w_i) + (1 - \alpha) * \mathrm{R}(p_i). \qquad (5)$$

A higher $\widetilde{w}_i$ indicates that candidate $\mathbf{C}_i$ excels in both pairwise scoring and the model's confidence. It ensures that both the winning score assigned by LLMs and the confidence of the model itself are jointly considered during the subsequent curation process.

**Preference Curation.** For each sample in the data pool, we curate the **preferred** and **non-preferred** candidates after deriving the confidence-weighted ranking scores $\widetilde{w}_i$ for all candidates. Specifically, the candidate with the highest $\widetilde{w}_i$ is selected as the preferred response $\mathbf{y}_w$ and the one with the lowest $\widetilde{w}_i$ is chosen as the non-preferred response $\mathbf{y}_l$.

### 3.4. Calibration via Preference Fine-tuning

To refine the model's rationales, we employ Direct Preference Optimization (DPO) (Rafailov et al., 2024), which encourages the model to favor preferred responses $\mathbf{y}_w$ over non-preferred ones $\mathbf{y}_l$ without relying on an explicit reward model. This calibration process adopts an **iterative** approach, progressively enhancing the model across multiple iterations. At each iteration $t$, the model $\pi_{t-1}$ from the previous iteration generates updated preference pairs $\mathcal{D}$ (with $t = 0$ corresponding to the seed model). These pairs guide the subsequent fine-tuning, where $\pi_t$ is calibrated by optimizing the log-sigmoid of the preference margin:

$$\mathcal{L}(\pi_t; \pi_{t-1}) = -\mathbb{E}_{(x, \mathbf{y}_w, \mathbf{y}_l) \sim \mathcal{D}}$$
$$\left[ \log \sigma \left( \beta \log \frac{\pi_t(\mathbf{y}_w \mid x)}{\pi_{t-1}(\mathbf{y}_w \mid x)} - \beta \log \frac{\pi_t(\mathbf{y}_l \mid x)}{\pi_{t-1}(\mathbf{y}_l \mid x)} \right) \right],$$
$$(6)$$

where $\sigma(\cdot)$ denotes the log-sigmoid function and $\beta$ is a scaling factor. By maximizing this margin at each iteration, the model learns to robustly favor the optimal RAP candidates, ensuring continual improvement in the process.

*Table 1.* **The main results of SRC on LLaVA-1.5 and LLaVA-Next across all benchmarks.** CP: Coarse Perception; FP: Fine-grained Perception; IR: Instance Reasoning; LR: Logical Reasoning; S&T: Science and Technology.

| Method | Overall | General Capabilities | | | | Math | S&T | Specific-domain VQA | | |
| --- | --- | --- | --- | --- | --- | --- | --- | --- | --- | --- |
| | | CP | FP | IR | LR | | | ChartQA[VAL] | LLaVA[W] | HallusionBench |
| **LLaVA-1.5-7B** | 32.1 | 59.2 | 25.2 | 39.6 | 24.4 | 24.0 | 20.4 | 17.8 | 61.8 | 24.7 |
| + POVID (Zhou et al., 2024a) | 32.2 | 58.0 | 25.2 | 40.0 | 24.4 | 23.6 | 22.0 | 18.4 | 65.9 | 24.8 |
| + HA-DPO (Zhao et al., 2023) | 33.0 | 58.0 | 26.0 | 40.0 | 28.0 | 24.8 | 21.2 | 17.7 | 65.8 | 26.2 |
| + SIMA (Wang et al., 2024c) | 32.8 | 59.6 | 26.4 | 40.8 | 24.4 | 27.2 | 18.4 | 18.6 | 62.2 | 24.1 |
| + SeVA (Zhu et al., 2024) | 33.3 | 58.0 | 28.4 | 40.8 | 24.8 | 25.6 | 22.0 | 18.4 | **66.3** | 27.1 |
| + RLAIF-V (Yu et al., 2024b) | 33.7 | 60.4 | 29.6 | 42.0 | 24.0 | 24.0 | 22.4 | 18.9 | 64.1 | 21.6 |
| + CSR (Zhou et al., 2024b) | 32.7 | 57.6 | 26.4 | 39.2 | 24.4 | 26.4 | 22.4 | 19.3 | 64.5 | 25.7 |
| **+ SRC (ours)** | **38.9** | **61.6** | **33.2** | **43.6** | **32.8** | **33.2** | **28.8** | **24.4** | 65.5 | **27.3** |
| **LLaVA-Next-8B** | 42.6 | 63.6 | 41.2 | 54.8 | 40.4 | 29.6 | 26.4 | 68.7 | 64.1 | 32.1 |
| **+ SRC (ours)** | **47.6** | **69.2** | **45.6** | **52.4** | **45.6** | **39.6** | **33.2** | **71.4** | **67.4** | **37.5** |

### 3.5. R-Scorer: An Scoring Model Tailored for SRC

**Motivation.** Our early experiments indicate that the scoring performance of open-source LLMs (Dubey et al., 2024; Yang et al., 2024), even with their 70B versions, is still unsatisfactory. While proprietary LLMs (*e.g.*, GPT-4o (OpenAI, 2024a)) manifest superior scoring performance, their costs render them impractical for large-scale pairwise scoring tasks. Thus, we developed a 1.5B model *tailored for scoring*, aiming to reduce reliance on proprietary LLMs while significantly improving the scoring efficiency.

**Data Collection and Training.** We sample 40K pairwise scoring examples from Section 3.2 and then sent them to GPT-4o (OpenAI, 2024a) to obtain scores. To improve the scoring quality and alleviate the scoring bias of GPT-4o, manual filtering and resampling are conducted, resulting in 21K training samples with an approximately normal distribution (details can be viewed in Appendix B.3). Using this curated scoring training set, we fine-tuned the Qwen-2.5-1.5B model (Yang et al., 2024) with LoRA and construct the R-Scorer model. Notably, during the scoring phase, we incorporate factual checks. However, for efficiency training, these checks were neglected in the training stage. Despite this, we observe that the model can still demonstrate robust generalization, effectively scoring candidates considering factual correctness.

## 4. Experiments

### 4.1. Setup

**Models.** We evaluate the effectiveness of the proposed SRC on the two baselines: LLaVA-1.5-7B (Liu et al., 2024a) and the LLaMA-3-based LLaVA-Next-8B (Liu et al., 2024b). During the *rationale fine-tuning* phase, the LoRA rank for LLaVA-1.5 and LLaVA-Next-8B is set to 4 and 32, respectively, while the training data remains identical for both models. In the *calibration* phase, the LoRA rank for both baseline models is fixed at 256. Notably, each iteration

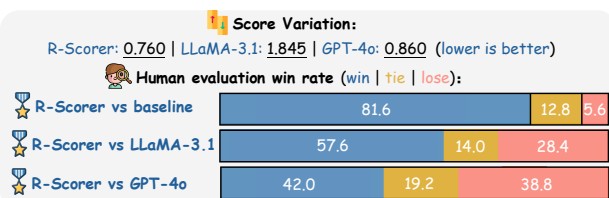

*Figure 7.* **Variation and Win Rate (Human Evaluation) of LLMs.** Variation is computed as the absolute difference. The evaluation samples are excluded during the training of R-Scorer.

utilizes the same data samples as in the previous iteration. More training details can be found in Appendix B.1.

**Datasets.** We sampled and augmented 57K VQA examples (refer to Section 3.5) to construct the training datasets of our paper. For the *perception & world knowledge*, we utilized datasets including LLaVA-150k (Liu et al., 2024c), VQAv2 (Goyal et al., 2017), ShareGPT-4V (Chen et al., 2025), GQA (Hudson & Manning, 2019), IDK (Min et al., 2024), TallyQA (Acharya et al., 2019), VizWiz (Gurari et al., 2018), and OODVQA (Tu et al., 2023). For *chart understanding*, we selected ChartQA (Masry et al., 2022) and DocVQA (Mathew et al., 2021). For *math & science*, we sampled from MathVision (Wang et al., 2024a) and AI2D (Kembhavi et al., 2016). The specific number of samples for each set is detailed in Appendix B.2. For the calibration process, we constructed a training dataset of 12k samples using the remaining data from the rationale fine-tuning phase, adhering to a 2:1:1 sample ratio.

**Evaluation.** We adopt MMStar (Chen et al., 2024c), an advanced benchmark of evaluation the comprehensive capabilities of LVLMs, as the primary evaluation metric. MMStar integrates multiple benchmarks, *e.g.*, MMBench (Liu et al., 2025), SEEDBench (Li et al., 2023a), MathVista (Lu et al., 2023), and MMMU (Yue et al., 2024), while addressing the data leakage issues and following the vision-centric QA principle. Additionally, we evaluate performance on specific-domain VQA tasks, which include ChartQA (Masry

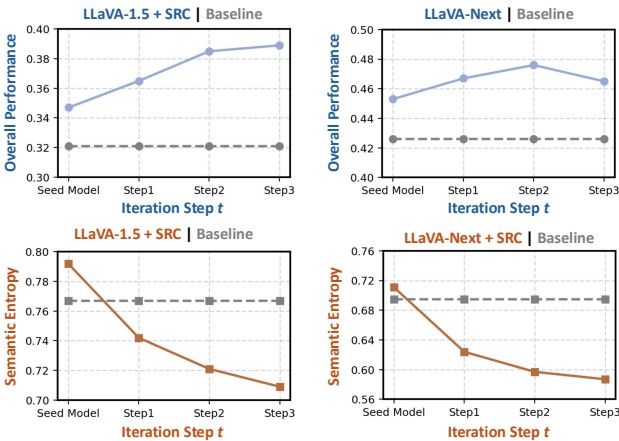

*Figure 8.* **Impact of SRC across iteration steps** on overall performance (MMStar) and semantic entropy for baseline models.

et al., 2022), LLaVABench (Liu et al., 2024c), and HallusionBench (Guan et al., 2023).

## 4.2. Main results

**SRC enhances both reasoning and perception capabilities of LVLMs across diverse tasks.** By aligning rationales with answers through rationale calibration, SRC significantly boosts overall performance on MMStar, improving LLaVA-1.5-7B and LLaVA-Next-8B from **32.1** and **42.6** to **38.9** and **47.6**, respectively. Beyond reasoning improvements in domains such as math and science, we can note that ***SRC also boosts perceptual abilities***, *e.g.*, with fine-grained perception in LLaVA-1.5 rising from **25.2** to **33.2** and coarse perception in LLaVA-Next increasing from **63.6** to **69.2**. Compared to existing post-training methods using DPO, SRC outperforms in most evaluation metrics, except for a minor shortfall on LLaVA-Bench. This is likely due to the adjustments in response formatting in SRC, affecting its "LLM-as-judge" evaluations. Furthermore, we tested ChartQA and Hallusionbench datasets, SRC also elevates specific-domain VQA scores. For instance, on LLaVA-Next, SRC increased the scores from **68.7** to **71.4** and from **32.1** to **37.5**, respectively. Figure 9 illustrates QA cases across different scenarios, demonstrating that models enhanced with SRC can better comprehend visual information and provide a reasonable rationale to generate accurate responses. When encountering hallucination-inducing cases, the SRC-enhanced models can accurately output image-grounded facts, producing correct judgements and responses.

**SRC exhibits a consistent enhancement in both capabilities and semantic consistency over iterations.** As shown in Figure 8, starting from the rationale fine-tuned seed model, LLaVA-1.5 and LLaVA-Next achieved initial MMstar scores of **34.7** and **42.6**, respectively, which were eventually increased to **38.9** (*step 3*) and **47.6** (*step 2*). By tracking semantic entropy (Farquhar et al., 2024), we ob-

*Table 2.* **Comparision of scoring strategies and models.** The complete evaluation results can be viewed in Table 5.

| Strategy / Models (1 iteration) | Overall | CP | FP | Math |
|---|---|---|---|---|
| Seed Model (LLaVA-1.5 baseline) | 34.7 | 57.6 | 26.0 | 28.4 |
| Score by count + R-Scorer | 35.4 | 58.0 | 20.8 | 29.2 |
| Score by sum (w/o fact check) + R-Scorer | 35.2 | 59.2 | 26.4 | 30.4 |
| **Score by sum + R-Scorer (default)** | **36.5** | **59.2** | **33.6** | **30.0** |
| Score by sum + Qwen-2.5-70B | 36.3 | 59.2 | 32.0 | 29.2 |
| Score by sum + LLaMA-3.1-70B | 36.1 | 58.8 | 31.6 | 28.8 |

*Table 3.* **Ablation of CoT prompting and rationale fine-tuning.** *: LLaVA-1.5 failed to test due to limited in-context learning.

| Ablation | Overall | CP | FP | Math |
|---|---|---|---|---|
| **LLaVA-1.5-7B** | 32.1 | 59.2 | 25.2 | 24.0 |
| + Prompt control* | - | - | - | - |
| + SFT *w/o* rationale | 33.7 | 59.0 | 24.4 | 29.2 |
| + SRC *w/o* Rationale Fine-tune | 34.0 | 59.6 | 24.8 | 28.8 |
| **+ SRC (default)** | **38.9** | **61.6** | **33.2** | **33.2** |
| **LLaVA-Next-8B** | 42.6 | 63.6 | 41.2 | 29.6 |
| + Prompt control | 40.8 | 62.4 | 37.2 | 35.6 |
| + SFT *w/o* rationale | 42.7 | 62.0 | 38.0 | 35.2 |
| + SRC *w/o* Rationale Fine-tune | 44.2 | 62.8 | 43.2 | 31.2 |
| **+ SRC (default)** | **47.6** | **69.2** | **45.6** | **39.6** |

served a slight increase in semantic entropy after rationale calibration compared to the baseline model. However, as the iterative process progresses, the semantic entropy of the model gradually decreases, ***reflecting enhanced semantic consistency in the model's output space and reduced response uncertainty***. We reckon that this improvement can be attributed to SRC's iterative candidate search and scoring process, during which the pair-wise scoring with R-Scorer identifies an response with optimal semantics (*i.e.*, answers with rationales). This process enables the model to progressively align the rationale and answer, leading to responses that exhibit consistent semantics across iterations.

**The R-Scorer manifests alignment with human preferences and lower sensitivity of candidate order.** Following the same procedure as in the training phase, we generated 200 sets of bidirectional scoring results, calculated variations after swapping candidate orders, and manually assessed the alignment with human preference ratings. As shown in Figure 7, the results demonstrate that our 1.5B R-Scorer outperforms the original 1.5B Qwen-2.5 (81.6% win rate) and the 70B LLaMA-3.1 (Dubey et al., 2024) (57.6% win rate) in terms of variation and alignment with human preferences. Compared to the proprietary model, GPT-4o, the R-Scorer ***achieves comparable performance*** (42.0% win rate) while offering significant advantages in inference efficiency and costs. Additional experimental details and comparative results are provided in Appendix C.

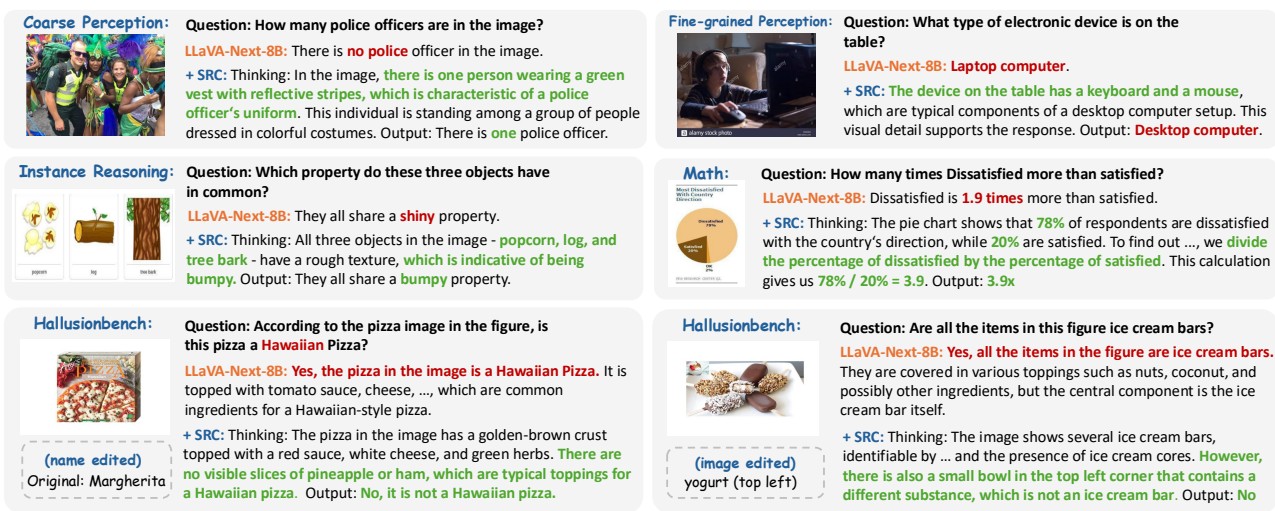

*Figure 9.* **Various QA scenarios**. The baseline model of LLaVA-Next-8B outputs incorrect responses. After incorporating SRC, the model not only provides correct answers but also *aligns the rationale effectively with the answers*.

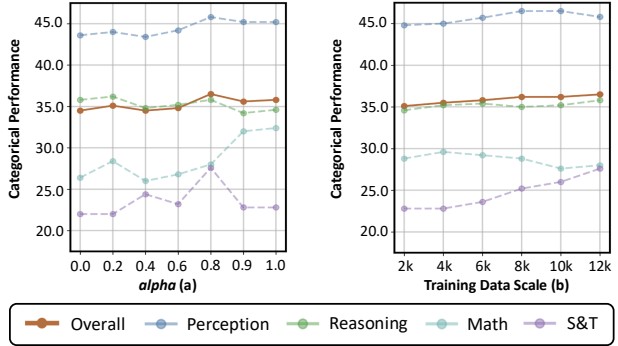

*Figure 10.* **Ablation of $\alpha$ and data scale of calibration process.** LLaVA-1.5 is used as our experiment baseline.

### 4.3. Ablation Study

**Different Scoring Strategies.** Table 2 examines the different scoring strategies using LLaVA-1.5 as our baseline. As observed, the winning score computed by the "sum" strategy (Equation 1) is more effective than that based on the "count" strategy (Equation 2). This suggests that the differences among candidates are challenging to distinguish between them using the "count" strategy. Moreover, incorporating factual checks during the scoring process proves to be beneficial, as it significantly *enhances the model's fine-grained perception capability* (from **26.4** to **33.6**). This demonstrates the model's ability to perform visual factual check by itself, thereby guiding the R-Scorer to assign more accurate scores and promote the calibration process.

**Different Scoring Models.** In Table 2, we compared the results of SRC using Qwen-2.5-70B (Yang et al., 2024) and LLaMA-3.1-70B (Dubey et al., 2024) in the pairwise scoring process. We can observe that SRC with R-Scorer

achieves the best post-training performance (from **34.7** to **36.5**). It reveals that while the R-Scorer has **only 2%** of the parameters compared to the former LLMs, it manifests superior scoring performance and provides better preference pairs for the calibration process.

**Prompt Control and Calibration without Rationales.** Table 3 presents an analysis of the prompting strategy and rationale learning. For conducting prompting in evaluation (details in Appendix B), while LLaVA-Next demonstrated improvements in mathematical reasoning, its overall performance exhibited a decline. Regarding rationale learning, we assessed the model's performance after fine-tuning using the same data with the original GT. Compared to the baseline models, the overall improvement was marginal, indicating that learning output rationales can enhance the model's performance *rather than data with GT itself*. We also tested SRC without the rationale fine-tuning stage, and the model still achieved a certain level of improvement (**32.1** to **34.0**). However, the improvement was less pronounced compared to calibrating the rationale fine-tuned seed model (**34.7** to **38.9**). This highlights that calibrating the model on rationales is more effective in exploring the models' potential than calibration based solely on responses.

**Tradeoff on $\alpha$ and the data scale of calibration.** As shown in Figure 10(a), we analyze the impact of $\alpha$ in the confidence-weighted winning score (Equation 5). Increasing $\alpha$ (*i.e.*, giving more weight to the R-scorer) improves overall performance, with the most notable gain in Math solutions (**26.4** to **32.4**), indicating that providing a rationale before generating an answer can enhance the model's performance to some extent. Figure 10(b) explores the impact of SRC training data scale, showing consistent performance gains, particularly in perception and science & technology

tasks, though with a slight decline in math solutions. These results highlight the tradeoffs between $\alpha$ and data scale in balancing task-specific performance and overall robustness.

## 5. Conclusion

In this paper, we introduce Self-Rationale Calibration (SRC) to address the alignment of rationales with answers in LVLMs. By iteratively enhancing models through rationale-oriented preference fine-tuning, SRC bridges the gap between reasoning processes and factual correctness in responses, particularly in VQA scenarios. We introduce R-Scorer, an efficient LLM-based scoring model, to assess candidate responses with pairwise comparisons, enabling nuanced evaluation of rationale quality. Extensive experiments demonstrate that SRC significantly improves both general and domain-specific capabilities, including reasoning and perception, outperforming existing post-training methods. Notably, SRC enhances semantic consistency and reduces uncertainty in responses across iterations. These results underscore the potential of rationale-based alignment to foster comprehensive improvements in LVLMs, advancing their reasoning and generalization capabilities.

## Impact Statement

This paper presents work whose goal is to advance the field of Machine Learning. There are many potential societal consequences of our work, none of which we feel must be specifically highlighted here.

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

# Appendix for Self-Rationale Calibration

In this appendix, we provide the details omitted in the main text, offering additional analyses and discussions.

- Appendix A: Additional implementation details of SRC (cf. Section 3 of the main paper).
- Appendix B: Additional experimental settings in this paper (cf. Section 4 of the main paper).
- Appendix C: Additional experiment results and discussions in this paper (cf. Section 4 of the main paper).

## A. Implementation Details

### A.1. Data Construction

In SRC, we adopt `Qwen2-VL-72B` (Wang et al., 2024b) to enhance our collected data samples by incorporating rationales (visual clues) before the answers. This approach enables the rationale fine-tuned model to autonomously generate rationales without requiring explicit prompts. ***The prompts used for data augmentation and filtering are illustrated in*** Figure 13. During data augmentation, we employ the tags `<perception>` and `<output>` as special identifiers to distinguish the rationale from the answer. In-context learning is achieved by providing a manually annotated sample case, which serves as the most critical step in the process. Subsequently, during rationale fine-tuning, these tags are replaced with "`Thinking:`" and "`Output:`", while keeping the original question unchanged. This design choice is based on our early experiments, which revealed that using placeholders with special symbols like `<..>` resulted in inferior model performance compared to natural language expressions such as "`Thinking:`" and "`Output:`". This finding is further discussed in Appendix C.3. Following this, we employ `Qwen-2.5-72B` (Yang et al., 2024) to filter the augmented data. The filtering criteria primarily evaluate whether the rationale is logical and fully supports the answer. If the rationale fails to effectively support the final answer, it is excluded from the dataset. More samples can be viewed in Figure 14.

### A.2. Candidate Generation

SRC employs a sentence-level beam search to generate diverse response candidates while ensuring inference efficiency. Unlike standard beam search, which constructs the search tree at the token level, ***SRC builds the search tree at the sentence level***, where each leaf node corresponds to a complete sentence rather than a token. For the first layer of the search tree, we expand the beam width to include five nodes (*i.e.*, `number of beams = 5`), and for other layers, we randomly sample two nodes per layer to maintain diversity while improving inference efficiency. The `eos_token_id` represents the token for a period, with its value varying depending on the underlying LLM, while the maximum sentence length is set to 1024 tokens. To avoid infinite loops, we implement mechanisms to eliminate sentence repetition and filter out undesired keywords, with each sentence limited to a maximum of 100 tokens. Additionally, the diversity penalty is set to 3.0 for LLaVA-1.5 and 2.0 for LLaVA-Next, balancing the diversity and quality of the sampled candidates.

### A.3. Pairwise Candidate Scoring

**Why use Pairwise Candidate Scoring?** A straightforward approach to scoring response candidates is to assign independent, absolute scores to each response. However, independent scoring has limitations both methodologically and in implementation. Methodologically, ***rationales differ from direct answers in that their semantics are less explicit and are inherently more open-ended***. This makes it difficult for independent scoring to capture the relative differences between rationales, as it lacks the ability to determine whether one candidate is better or worse compared to others. From an implementation perspective, we observed that ***open-source LLMs exhibit noticeable scoring biases when assigning absolute scores*** (see Figure 11). For instance, certain models, such as `GPT-4o-mini`, tend to cluster scores around 1 and 4, whereas others, like `GPT-4o`, skew towards 0 and 5. These biases complicate the preference curation process, as many candidates end up with identical scores, making it challenging to distinguish between preferred and non-preferred responses. To evaluate this approach, we further conducted a detailed analysis of the model's performance, as provided in Appendix C.1.

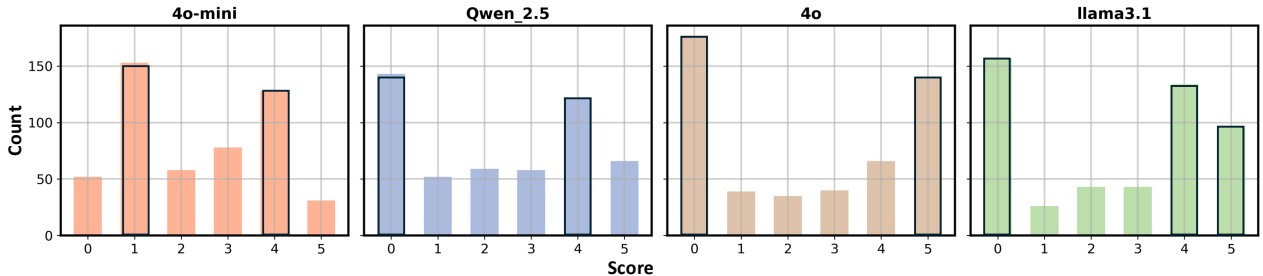

*Figure 11.* **The distributions of independent scoring through recent advanced LLMs**. The scoring prompt is detailed in Figure 16.

**Why use LLMs instead of MLLMs for scoring?** We opted for LLMs over MLLMs in the scoring process due to findings from prior research (Chen et al., 2024a), which highlighted the ***limitations of MLLMs in evaluative tasks*** (such as scoring and batch ranking) compared to LLMs. These limitations make LLMs a more reliable choice for our scoring strategy.

**Implementation.** The scoring prompt is presented in Figure 15. For LLMs, the evaluation of response candidates focuses on two key aspects: (1) ***the correctness of visual elements*** (*i.e.*, the alignment between vision and text), which is achieved through factual verification, and (2) ***the coherence between the rationale and the answer***, leveraging the "LLM-as-a-judge" capabilities. The scoring scale ranges from $-5$ to $5$. During our experiments, we observed notable variations in scores depending on the order of the candidates. To mitigate this issue, we implemented bidirectional scoring, as elaborated in the main paper. Additionally, we incorporated a scoring case into the prompt to enhance scoring stability through in-context learning. Once the scoring matrix for each sample was obtained, we calculated the absolute scores for each candidate to derive confidence-weighted winning scores, which were subsequently used for preference curation.

# B. Experiment Details

## B.1. LoRA in Rationale and Preference Fine-tuning

During the rationale calibration stage, the LoRA ranks for LLaVA-1.5 and LLaVA-Next are set to 4 and 32, respectively, with their corresponding LoRA learning scales set to twice the rank. The learning rates for both models are set to 1e-5. In the preference fine-tuning stage, the LoRA rank for both LLaVA-1.5 and LLaVA-Next is set to 256, with a learning scale of 512 and a learning rate of $5e - 7$. For DPO, the regularization weight $\beta$ is set to 0.1. Additionally, we incorporate an SFT loss following RPO (Liu et al., 2024d), with the loss weight for the SFT term set to 0.02. The iterative training is carried out over three iterations.

## B.2. Data Construction of SRC

We curated a training set of 57K samples from 11 popular datasets in LVLMs, encompassing three major categories: perception & world knowledge, chart understanding, and math & science. Detailed descriptions of each dataset and their respective contributions to the SRC training set are provided in Table 4. Before data augmentation, we applied rigorous filtering to remove general knowledge that does not require visual context for answering, which is often found in datasets like LLaVA-150k and ShareGPT-4V. This ensured that only QA pairs strongly tied to visual content were retained. After augmentation (Section 3.1), the final dataset comprised 43K samples, with examples shown in Figure 14. For the rationale fine-tuning and DPO calibration phases, we prioritized non-redundant entries and maintained a 2:1:1 ratio across the three major categories. Within each category, uniform sampling was employed to ensure diverse coverage of all data types.

## B.3. Data Construction of R-Scorer

The training of R-Scorer involves learning a pair-scoring process. We sampled 40K pairwise comparison examples generated by LLaVA-1.5, selecting candidates across different categories in a 2:1:1 ratio. To enhance the model's scoring sensitivity to the order of candidates, each pair of candidates was swapped, resulting in two samples for the R-Scorer training dataset (essentially incorporating 20K unique pairs of candidates). These samples were scored using GPT-4o-11-20, followed by manual curation to filter out anomalies—such as samples with high scoring variance after swapping candidate order—and resampling using a normal distribution. This process resulted in a final training set of 21K examples, as shown in Figure 19. Notably, the training process does not involve explicit factual checking. However, our experiments demonstrate that the model can generalize to consider factual correctness effectively.

| Category | Dataset | Description | # of items |
|---|---|---|---|
| Perception & World Knowledge | LLaVA-150k (Liu et al., 2024c) ShareGPT-4V (Chen et al., 2025) | Focuses on multimodal conversations generated by GPT-4V for advancing VQA and related tasks | 7155 |
| | GQA (Hudson & Manning, 2019) | Compositional question answering over images with reasoning and relational understanding | 5574 |
| | IDK (Min et al., 2024) | Explores uncertainty and knowledge gaps in visual question answering tasks | 2842 |
| | TallyQA (Acharya et al., 2019) | Centers on counting-based reasoning tasks across diverse image domains | 2980 |
| | VizWiz (Gurari et al., 2018) | Assists visually impaired users with real-world images and questions | 1993 |
| | OODVQA (Tu et al., 2023) | Addresses out-of-distribution generalization challenges in VQA | 667 |
| Chart Understanding | ChartQA (Masry et al., 2022) | Focuses on interpreting graphical data through question answering | 7761 |
| | DocVQA (Mathew et al., 2021) | Requires understanding textual and structural information in scanned documents | 4734 |
| Math & Science | MathVision (Wang et al., 2024a) | Combines symbolic and visual reasoning to solve mathematical problems | 1443 |
| | AI2D (Kembhavi et al., 2016) | Advances diagram understanding through visual and textual question answering | 7887 |

*Table 4.* **Overview of datasets sampled in SRC**. It includes three categories and 11 datasets commonly used in training of LVLMs.

| Strategy / Models (1 iteration) | Overall | CP | FP | IR | LR | Math | S&T |
|---|---|---|---|---|---|---|---|
| Seed Model (LLaVA-1.5 baseline) | 34.7 | 57.6 | 26.0 | 45.6 | 28.0 | 28.4 | 22.8 |
| Independent scoring + Qwen-2.5-72B | 35.3 | 58.4 | 27.2 | 40.0 | 32.4 | 31.2 | 22.4 |
| Score by count + R-Scorer | 35.4 | 58.0 | 20.8 | 48.0 | 32.0 | 29.2 | 24.4 |
| Score by sum (w/o fact check) + R-Scorer | 35.2 | 59.2 | 26.4 | 40.4 | 33.6 | 30.4 | 21.2 |
| Score by sum + R-Scorer-7B | 36.8 | 59.6 | 32.8 | 40.4 | 33.2 | 31.2 | 23.6 |
| **Score by sum + R-Scorer-1.5B (default)** | **36.5** | **59.2** | **33.6** | **40.8** | **32.8** | **30.0** | **22.8** |
| Score by sum + Qwen-2.5-72B | 36.3 | 59.2 | 32.0 | 40.8 | 31.6 | 29.2 | 25.0 |
| Score by sum + LLaMA-3.1-70B | 36.1 | 58.8 | 31.6 | 41.2 | 31.2 | 28.8 | 25.0 |

*Table 5.* **Comparison of scoring strategies and models.** The evlauation benchmark is MMStar (Chen et al., 2024c). CP: Coarse Perception, FP: Fine-grained Perception, IR: Instance Reasoning, LR: Logical Reasoning, S&T: Science and Technology.

## B.4. Prompt for Baseline Models in Evaluation

In Table 3, we also evaluated the performance of the original baseline models (LLaVA-1.5 and LLaVA-Next) using a rationale prompt designed to achieve "providing a rationale before answering". The prompt used during the experiments was as follows: "`<image><question>` Please think step-by-step and follow the output format: Thinking: xxx Output: xxx". For LLaVA-1.5, we found that this prompt consistently failed to elicit a rationale from the model, likely due to the weaker LLM capabilities of Vicuna in LLaVA-1.5. In contrast, LLaVA-Next (LLaMA-3) was able to generate rationales using the rationale prompt. From Table 3, we observe that while this approach improved the model's performance on math-related solutions, it led to a decline in performance on other types of tasks, such as perception and logical reasoning.

# C. More Experiments and Discussions

## C.1. Comparison of Scoring Strategies and LLMs

In this section, we present the results of scoring strategies on improving model performance during the calibration process. We also provide a comparative analysis of independent scoring and experiments with larger-scale R-Scorers.

**Independent Scoring.** In this scoring process, the LLM assigns scores to each response candidate independently and then

| Ablation | Overall | CP | FP | IR | LR | Math | S&T |
|---|---|---|---|---|---|---|---|
| **LLaVA-1.5-7B (baseline)** | 32.1 | 59.2 | 25.2 | 39.6 | 24.4 | 24.0 | 20.4 |
| (i) without special tag | 34.2 | 59.6 | 26.4 | 41.6 | 26.8 | 26.0 | 24.8 |
| (ii) \<thinking\>+\<output\> | 29.4 | 55.6 | 19.4 | 38.0 | 20.8 | 23.2 | 19.6 |
| (iii)[thinking]+[output] | 31.3 | 57.2 | 21.8 | 38.4 | 25.2 | 25.2 | 20.0 |
| **(iv) Thinking+Output (default)** | **34.7** | **57.6** | **26.0** | **45.6** | **28.0** | **28.4** | **22.8** |

*Table 6.* **Evaluation of response format in rationale fine-tuning.** The evlauation benchmark is MMStar (Chen et al., 2024c). CP: Coarse Perception; FP: Fine-grained Perception; IR: Instance Reasoning; LR: Logical Reasoning; S&T: Science and Technology.

| Ablation | Overall | CP | FP | IR | LR | Math | S&T |
|---|---|---|---|---|---|---|---|
| **LLaVA-1.5-7B** | 32.1 | 59.2 | 25.2 | 39.6 | 24.4 | 24.0 | 20.4 |
| + Prompt control | - | - | - | - | - | - | - |
| + SFT *w/o* rationale | 33.7 | 59.6 | 24.4 | 40.4 | 26.0 | 29.2 | 22.8 |
| + SRC *w/o* Rationale Fine-tune | 34.0 | 60.0 | 28.8 | 41.2 | 31.6 | 28.8 | 23.6 |
| **+ SRC (default)** | **38.9** | **61.6** | **33.2** | **43.6** | **32.8** | **33.2** | **28.8** |
| **LLaVA-Next-8B** | 42.6 | 63.6 | 41.2 | 54.8 | 40.4 | 29.6 | 26.4 |
| + Prompt control | 40.8 | 62.4 | 37.2 | 52.0 | 35.2 | 35.6 | 22.4 |
| + SFT *w/o* rationale | 42.7 | 62.0 | 38.0 | 53.2 | 38.4 | 35.2 | 27.6 |
| + SRC *w/o* Rationale Fine-tune | 44.2 | 62.8 | 43.6 | 54.0 | 47.6 | 31.6 | 27.6 |
| **+ SRC (default)** | **47.6** | **69.2** | **45.6** | **52.4** | **45.6** | **39.6** | **33.2** |

*Table 7.* **Ablation of CoT prompting and rationale fine-tuning.** The evlauation benchmark is MMStar (Chen et al., 2024c). CP: Coarse Perception; FP: Fine-grained Perception; IR: Instance Reasoning; LR: Logical Reasoning; S&T: Science and Technology.

selects preferred and non-preferred candidates for iterative calibration. The prompt template used is illustrated in Figure 16, where the pairwise scoring-related components are removed while retaining the "*question*", "*response candidate*", "*factual check*", and "*reference answer*". The experimental results are shown in Table 5, where we observe that with the same scoring model (Qwen-2.5-72B), *the pairwise scoring strategy employed in SRC significantly outperforms the independent scoring strategy*. The improvement is particularly notable in fine-grained perception capability (+6.4).

**Larger R-Scorer.** Using Qwen-2.5-7B (Wang et al., 2024b) as the base LLM, we performed LoRA fine-tuning (rank=16) with consistent scoring training data, as detailed in Section 3.5. The results indicate that *a larger R-Scorer provides overall better calibration performance*, particularly in categories such as reasoning, mathematics, and science & technology. We attribute this improvement to the larger LLM's enhanced reasoning and world knowledge capabilities, enabling it to better capture rationale differences in these areas and select superior preference pairs for calibration. However, considering the 4–5x inference burden, we choose *the 1.5B version of R-Scorer* in SRC for a more balanced trade-off.

## C.2. More results and visualizations

In this section, we provide additional experimental results and visualizations. The complete results of the ablation studies are shown in Table 7. We also present the scoring visualizations of the R-Scorer, as illustrated in Figure 18. Instead of directly outputting scores, the R-Scorer first provides a concise rationale before assigning a score. In Figure 17, we include additional responses from LLaVA-Next after applying SRC, further demonstrating the effectiveness of the model's rationale.

## C.3. Response Format in Rationale Fine-tuning

During the rationale fine-tuning phase, we introduced specific tags to assist the model in distinguishing between rationale and answer segments. Four variants were evaluated in our experiments: (i) no special tags, (ii) using "\<thinking\>" and "\<output\>" tags, (iii) employing "[thinking] and [output]" tags, and (iv) utilizing natural language-like expressions such as "Thinking and Output". The results, tabulated in Table 6, reveal that the model's performance significantly degraded after rationale fine-tuning when explicit special tags (variants ii and iii) were introduced. This indicates that special tags might disrupt the model's inherent logical structure during language generation, leading to decreased performance. In contrast, when no tags were applied (variant i) or natural language-style tags (variant iv) were used, the rationale effectively improved the quality of the answers. Remarkably, even without the subsequent rationale calibration process, the model demonstrated enhanced overall capabilities.

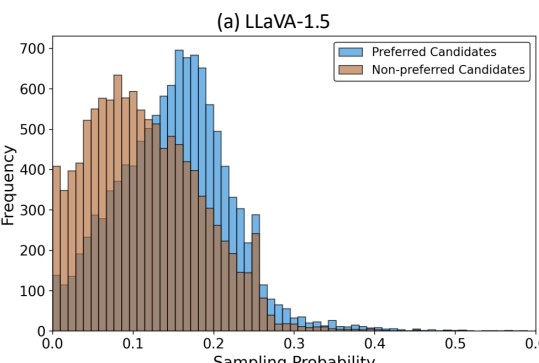 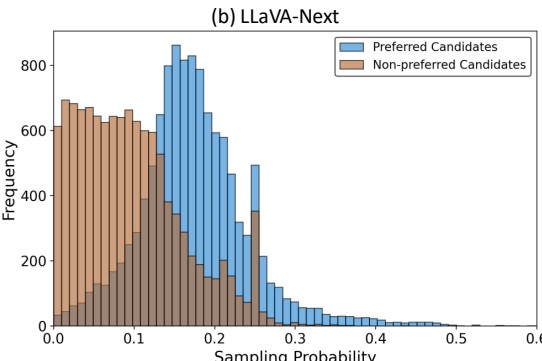

*Figure 12.* **Confidence distribution** of preferred and non-preferred candidates for LLaVA-1.5 (Liu et al., 2024a) and LLaVA-Next (Liu et al., 2024b). The sampling probability is computed by normalizing the logits of each candidate through the SoftMax function.

### C.4. The Confidence of Preferred and Non-preferred Candidates

One inevitable challenge in the scoring process is the presence of multiple candidates with similar quality, which may cause LLMs to assign neutral scores. Additionally, the inherent preference biases of LLMs can lead to suboptimal scoring outcomes. Thus, we incorporate the confidence of response candidates as a post-processing step to refine the scoring results. Here, we analyze ***the relationship between curated candidates and their confidence levels***. Our findings reveal that the preferred candidates from both LLaVA-1.5 and LLaVA-Next generally ***exhibit higher confidence*** compared to non-preferred candidates. The difference in confidence between preferred and non-preferred candidates is slightly more noticeable for LLaVA-Next. These results indicate a positive correlation between response candidates and their confidence scores; however, it is important to note that low confidence does not necessarily imply that a candidate is non-preferred.

**AUGMENT** 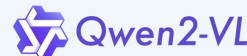

**Please augment each JSON-formatted dialogue by adding visual clues to {{"from": "gpt", "value": "xxxx"}} responses based on the user's question ({{"from": "human"}}).**

**# Instructions:**
1. Add visual clues in <perception> tags for each {{"from": "gpt", "value": "xxxx"}} response. These clues should support the final answer by first identifying the relevant objects and then providing reasoning based on the visual clues.
2. Do not modify the original "value": "xxxx". It must be enclosed within <response> tags.

**# Rules (the content in the <perception> tag):**
1. The visual clues should provide reasoning in a natural flow, first locating the objects, then identifying key features relevant to the question, and finally supporting the answer based on these features.
2. Do not directly answer the question in the <perception> tag.
3. If the question does not require visual clues to answer, output "No perception required." in the tag.

**# Sample:**
```
{
    "id": "xxxx",
    "image": "coco/train2017/xxxx.jpg",
    "conversations": [
        {
            "from": "human",
            "value": "<image_1>\nHow many giraffes are in the image?"
        },
        {
            "from": "gpt",
            "value": "<perception>In the image, a giraffe is clearly visible on the left side, with its long neck and spotted coat
standing out against the background. Another giraffe can be seen in the middle of the frame, its full body in view and slightly
taller than the first. Both giraffes are easily distinguishable based on their unique patterns and long necks.</perception>
<response>There are two giraffes in the image.</response>"
        }
    ]
}
```

**# Input: {input}**

**FILTER**

You will receive a JSON of a multimodal model conversation.
**Your task is to check the validity of the visual reasoning path in the <perception> tag restrictly.**

**# Guideline：**
* Valid reasoning path: The model should be able to reach the response in the <response> tag through the visual reasoning path in the <perception> tag. The reasoning path must be logical and fully support the response in the <response> tag.
* Invalid reasoning path: The path does not lead to the result in the <response> tag or does not provide any helpful information for the response in the <response> tag.

**# Output：**
* If the reasoning path is valid, return 'valid'.
* If the reasoning path is invalid, return 'invalid' with a brief reason in parentheses.

Now, please check: **{input}**

*Figure 13.* **The prompt for augmenting the VQA samples and filtering low-quality augmented samples**. For different VQA datasets (*e.g.*, perception and math solution), we manually design different samples for in-context learning.

**Perception:**

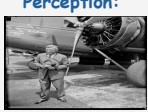

**Question: Is this man skiing in the trees?**

**Thinking:** The skier is on a clear path without any immediate obstacles such as trees, suggesting he is not skiing through the trees.
**Output:** No

**Perception:**

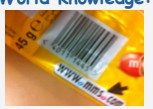

**Question: What is the red letter on the plane?**

**Thinking:** The image is in black and white, making it impossible to determine the color of any objects or letters on the plane. **Output:** There is no red letter on the plane in the picture as it's a black and white photo.

**World Knowledge:**

**Question: Is this Bibsy?**

**Thinking:** The image shows a yellow package with a barcode and some text, including 'www.mms.com'. There is no indication or visual clue that suggests this is related to Bibsy. **Output:** No, this is not Bibsy.

**Chart:**

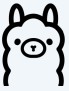

**Question: What is the first data in orange line?**

**Thinking:** The first data point on the orange line, which represents U.S. confidence in Putin, is located at the far left of the graph. This point corresponds to the year 2003 and has a value of 41%, indicating that 41% of Americans had confidence in Putin at that time.
**Output:** 41%

**Math:**

**Question: The diagram shows an equilateral triangle and a regular pentagon. What is the value of $x$?**

**Thinking:** The diagram illustrates an equilateral triangle and a regular pentagon sharing a common vertex. The interior angle of a regular pentagon is 108 degrees, and since the triangle is equilateral, each of its angles measures 60 degrees. The value of x can be found by subtracting the interior angle of the pentagon from the straight angle formed at the shared vertex, which is 180 degrees. Therefore, x = 180 - 108. **Output:** Answer is 132.

*Figure 14.* **The augmented QA data samples used in SRC.** The samples are augmented by Qwen2-VL-72B (Wang et al., 2024b).

---

## Pair-wise Scoring

You are a precise evaluator of model response quality.

Each Response has two part, i.e., "Thinking" part (Thinking: xxx) and "Output" part (Output: xxx). Your task is to compare Response A and Response B based on the provided question, their corresponding factual checks, and the reference (golden) answer.

**# Please evaluate from the following perspectives:**
0. The **top** priority: The Output of a good response MUST be aligned with the Output of the reference answer.
1. Whether the response's "Thinking" part contains content that is counter-intuitive to the image facts (for example, outputting something that does not exist).
2. Whether the response's "Thinking" part can effectively support the "Output" part.

**# Score criteria:**
- The comparison score ranges from **-5 to 5**. The sign of the score MUST clearly reflect which response is superior: Positive score (+): A > B | Negative score (-): B > A
- If Response A is better than Response B, MUST assign a **positive** score (closer to 5).
- If Response B is better than Response A, MUST assign a **negative** score (closer to -5).

**# Return**
Please first evaluate the response in ONLY one sentence and return the final score at the end. The score is wrapped with the <> symbol (for example, <5>, <-5>).

**# Example:**
(A is better than B => MUST give a **positive** score): Response A describes xxx correctly, which is visible in the image, while Response B inaccurately describes counter-factual xxxx. Therefore, Response A is more accurate and reliable. <X>
(B is better than A => MUST give a **negative** score): Response A incorrectly describes xxx, which is not visible in the image, while Response B accurately describes xxxx without any counter-intuitive content. Therefore, Response B is more accurate and reliable. <-X>

---
**Now please make an evaluation. The image corresponding question is: {question}**
---
**Response A: {response_a}**
**Factual Check of Response A: {check_a}**
---
**Response B: {response_b}**
**Factual Check of Response B: {check_b}**
---
**Reference Answer: {reference_answer}**

*Figure 15.* **The Pairwise Scoring prompt** adopted in Section 3.2 for LLMs.

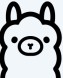

**Independent Scoring**

You are a precise evaluator of model response quality.

Each Response has two part, i.e., "Thinking" part (Thinking: xxx) and "Output" part (Output: xxx). Your task is to compare Response A and Response B based on the provided question and the reference (golden) answer.

**# Please evaluate from the following perspectives:**
0. The **top** priority: The "Output" of a good response MUST be aligned with the "Output" of the reference answer.
1. Whether the response's "Thinking" part contains content that is close to the "Thinking" of the reference answer.

**# Score criteria:**
- Score ranges from **0 to 5**. The sign of the score MUST clearly reflect which the above perspectives.
- A score of **0** means the response is very poor.
- A score of **5** means the response is excellent.

**# Return**
Please first evaluate the response in ONLY one sentence and return the final score at the end. The score is wrapped with the <> symbol (for example, <5>).

---
**Now please make an evaluation. The image corresponding question is: {question}**
---
**Response: {response_a}**
**Factual Check of Response A: {check_a}**
---
**Reference Answer: {reference_answer}**

*Figure 16.* **The independent scoring prompt** adopted in Appendix 3.2 using LLMs.

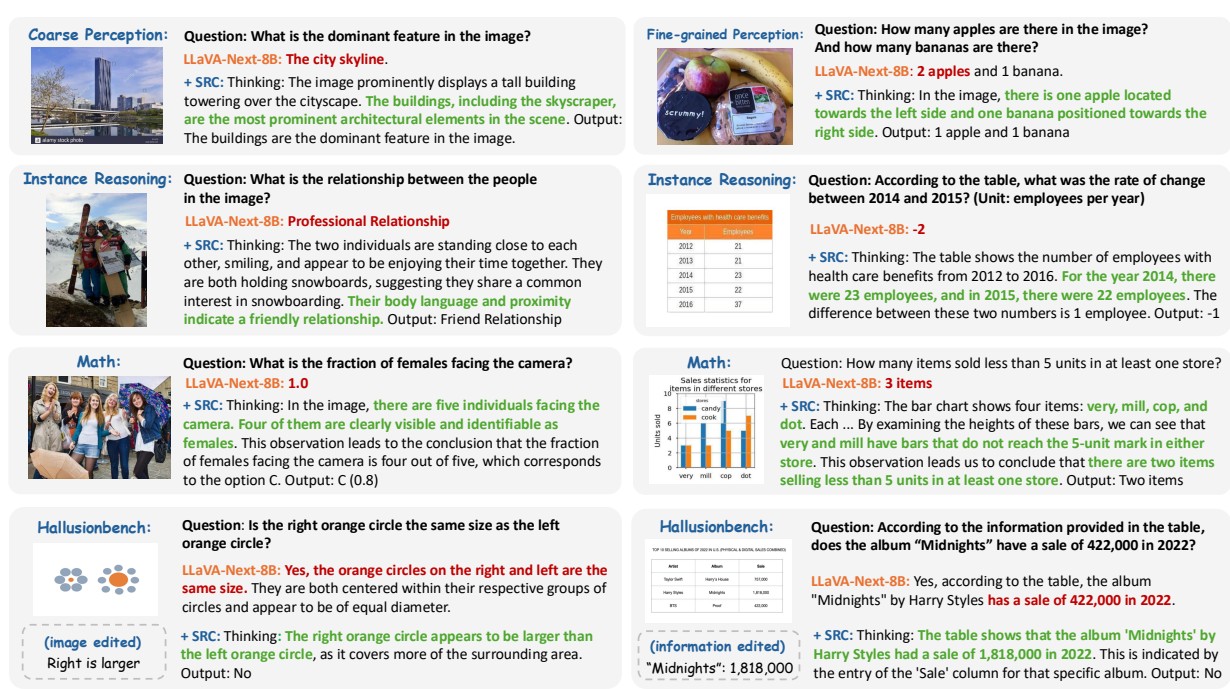

*Figure 17.* **Various QA scenarios**. The baseline model of LLaVA-Next-8B outputs incorrect responses. The cases are selected from MMStar (Chen et al., 2024c) and HallusionBench (Guan et al., 2023).

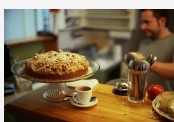

**Question: How many police officers are in the image?**

**R-Scorer:** Response A incorrectly identifies the coffee cake as a dessert, while Response B suggests it is for breakfast, which is closer to the reference answer's context of a morning or lunchtime meal, making Response B slightly more aligned with the reference. **<-2>**

**Candidate #1:** Thinking: A slice of coffee cake is placed on a glass plate, **which is typically used for serving desserts**. This suggests that the coffee cake is intended to be served as a dessert. Output: **Dessert**

**Candidate #2:** Thinking: The coffee cake is placed on a table, and there is a cup of coffee nearby, suggesting that it is meant to be enjoyed with a hot beverage. This combination is commonly associated with breakfast. Output: The coffee cake is intended to be served with coffee, **which is typically associated with breakfast.**

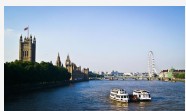

**Question: What prominent structure can be seen in the background besides the city buildings?**

**R-Scorer:** Response A provides a more detailed and accurate description of the Ferris wheel, aligning well with the reference answer and the factual checks, while Response B inaccurately describes the Ferris wheel as a white top. **<3>**

**Candidate #1:** In the background, there is a large structure with a distinctive shape and color, which is characteristic of a Ferris wheel. This structure stands out against the cityscape and the river. Output: **A Ferris wheel can be seen in the background of the image**, along with the city buildings and the river.

**Candidate #2:** Thinking: **A large structure with a distinctive white top** can be seen in the background, **towering above the cityscape**. This structure is easily recognizable due to its height and unique design. Output: **A prominent structure in the background besides the city buildings is the London Eye, a giant Ferris wheel.**

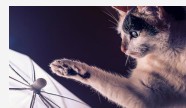

**Question: Is the cat spotted in the image?**

**R-Scorer:** Response A is more aligned with the reference answer as it correctly identifies the cat's fur pattern as spotted, while Response B incorrectly states that the black patches cover a significant portion of the cat's body. <3>

**Candidate #1:** Thinking: The image shows a cat with distinct black patches on its fur. The cat's fur pattern is clearly visible, indicating that it is indeed spotted. Output: **Yes, the cat is spotted in the image**.

**Candidate #2:** The image shows a cat with distinct black patches on its fur. These patches are clearly visible and **cover a significant portion of the cat's body**. Output: **Yes, the cat is spotted in the image**.

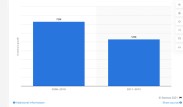

**Question: How much did investment in personalized medicine grow between 2006 and 2010?**

**R-Scorer:** Response A incorrectly identifies the growth percentage as 51%, while Response B also incorrectly identifies it as 77%; however, both responses fail to match the reference output of 73%. In terms of the \"Thinking\" part, Response B is closer to the reference by correctly identifying the relevant period, but it still provides an incorrect percentage. Therefore, Response B is slightly better in terms of understanding the context, but both responses fail in the output alignment. **<-1>**

**Candidate #1:** Thinking: The bar representing the year 2010 shows an increase in investment compared to the previous year. The height of the bar indicates the percentage change in investment. In this case, **the bar reaches up to 51%,** which is the percentage increase in investment during that period. Output: 51

**Candidate #2:** Thinking: The bar chart shows two bars representing the investment in personalized medicine from 2006 to 2015. For the period 2006-2010 **there is a bar that reaches up to 77% on the y-axis**, indicating the growth in investment during that time frame. Output: 77

*Figure 18.* **The scoring cases of R-Scorer.** Though the answers/visual elements are often correct (green), the rationales may be counterfactual (red) or insufficient (orange). The pair-wise scorer can capture subtle diffences between candidates.

**Input:**                                                      **Data Sample of Training R-Scorer**

You are tasked with evaluating the quality of model responses based on a given question and a reference (golden) answer. Each response consists of two parts:

**1.Thinking**: The reasoning process behind the response.

**2.Output**: The final answer provided by the model.

**Evaluation Criteria**

**1.Top Priority**: The "Output" of a good response must align with the "Output" of the reference answer.

**2.Thinking Alignment**: The "Thinking" part of the response should be close to the "Thinking" of the reference answer.

**Scoring System**

•**Score Range**: -5 to 5

- **Positive Score (+)**: Indicates that Response A is better than Response B.
- **Negative Score (-)**: Indicates that Response B is better than Response A.

•**Score Assignment**:

- If Response A is better, assign a **positive** score (closer to 5).
- If Response B is better, assign a **negative** score (closer to -5).

**Evaluation Example**

•**Positive Score Example**: "Response A describes xxx correctly, which is visible in the image, while Response B inaccurately describes counter-factual xxxx. Therefore, Response A is more accurate and reliable. <X>"

•**Negative Score Example**: "Response A incorrectly describes xxx, which is not visible in the image, while Response B accurately describes xxxx without any counter-intuitive content. Therefore, Response B is more accurate and reliable. <-X>"

**Current Evaluation Task**

**Question**: What color is the traffic light in the image?

**Responses**:

•**Response A**:

- **Thinking**: The image shows a traffic light with three distinct colors: red, yellow, and green. The red light is clearly visible, indicating that vehicles must stop.
- **Output**: The traffic light in the image is red.

•**Response B**:

- **Thinking**: The image shows a traffic light with three distinct colors: red, yellow, and green. The red light is clearly visible, indicating that vehicles must stop. The yellow light is partially obscured by the fog, but it is still visible enough to see its color. The green light, which is the signal for vehicles to proceed, is also partially obscured by fog but can still be seen.
- **Output**: The traffic light in the image has a red, yellow, and green light. The red light is currently showing, indicating that vehicles must stop. The yellow light is partially obscured by the fog, but it is still visible enough to see its color.

**Reference Answer**:

•**Thinking**: The traffic light in the image has three distinct sections, each with a different colored light. The top section is red, the middle section is yellow, and the bottom section is green. The yellow light is currently illuminated.

•**Output**: The traffic light in the image is displaying the yellow color.

---

**Output:**

Response A incorrectly identifies the traffic light as displaying red when the reference indicates it is displaying yellow, while Response B provides additional, albeit unnecessary, details but still misidentifies the illuminated color as red. Therefore, both responses fail to align with the reference, but Response B's expanded explanation of the traffic light structure slightly adheres more closely to the "Thinking" of the reference. <-2>

*Figure 19.* **Training samples of R-Scorer.** The factual check is not involved in the training stage.

