# OpenReview forum: "Towards Rationale-Answer Alignment of LVLMs via Self-Rationale Calibration"
_ICML.cc/2025/Conference — ICML 2025 poster_

### Official Review · Reviewer_CYZ4 · 2025-03-14

**Overall Recommendation:** 2

**Summary:**

The paper targets misalignment between rationales and answers in Large Vision-Language Models (LVLMs), particularly in VQA tasks. It introduces Self-Rationale Calibration (SRC), a framework that iteratively aligns rationales with answers using a combination of rationale fine-tuning, pairwise candidate scoring, and confidence-weighted preference curation.

The main contributions include (1) the SRC framework, which improves the factual consistency and reasoning quality of LVLMs, (2) a pairwise scoring strategy with R-Scorer, and (3) extensive experiments demonstrating significant performance improvements across multiple benchmarks

**Claims And Evidence:**

No serious flaws found

**Essential References Not Discussed:**

Related works are clear

**Experimental Designs Or Analyses:**

No serious flaws found

**Methods And Evaluation Criteria:**

See weakness

**Other Comments Or Suggestions:**

The paper excessively uses various special text styles, including bold, italics, colored text, etc., which make reading difficult and visually unpleasant. Additionally, the color of in-text citations and section references seem different from the standard template; I am not sure if this leads to formatting issues.

**Other Strengths And Weaknesses:**

The proposed method is reasonable and meaningful, with experimental results demonstrating its effectiveness across multiple datasets. However, the novelty of this work is questionable, as the core concept of Self-Rationale Calibration (SRC) does not introduce fundamentally new learning principles or new findings.
Specifically, the key innovation is shifting alignment focus from correctness alone to rationale consistency. Previous methods align answers or vision-text pairs, but SRC aligns the thinking process behind answers. The integration of rationale fine-tuning, pairwise scoring, and preference learning for multimodal alignment is reasonable. However, SRC primarily combines existing techniques (CoT, preference fine-tuning) rather than introducing new theoretical principles.

**Questions For Authors:**

Could the authors clarify the novelty beyond integrating existing techniques? In other words, is the contribution primarily on the engineering and empirical improvements side?

**Relation To Broader Scientific Literature:**

See Other Strengths And Weaknesses

**Theoretical Claims:**

No serious flaws found

---

> ### Author Rebuttal · Authors · 2025-03-31
>
> Dear reviewer, due to **space limits** of initial rebuttal, we are unable to elaborate on details or minor points, but we would be glad to clarify any further concerns in the next-round reply.
>
> ---
>
> > **The novelty and contribution of SRC.**
>
> We sincerely appreciate the reviewer’s positive feedback regarding the reasonableness and effectiveness of our proposed method. We would also like to respectfully **clarify the novelty and contribution of our work**, specifically addressing the reviewer’s concern regarding the originality of SRC beyond simply integrating existing techniques.
>
> **# Motivation and New Findings:**
>
> Existing LVLM post-training approaches primarily emphasize aligning outputs based on correctness or vision-text consistency. However, as we observed and illustrated through concrete examples (Figure 2), merely correct answers may result from spurious correlations rather than genuine understanding or reasoning processes. This observation **highlights a critical yet overlooked misalignment**—the disparity between a model’s rationale (its underlying reasoning) and the final answer.
>
> Our novel insight lies here: by explicitly calibrating the alignment between rationales and answers, SRC **significantly improves not only factual correctness but also logical consistency and semantic robustness**. We empirically validate this improvement via semantic entropy measurements (Section 4), showcasing clear benefits in scenarios that demand reliable and interpretable multimodal reasoning, such as visual question answering (VQA).
>
> **# Novelty and Contributions:**
>
> We respectfully emphasize that **SRC is more than merely an engineering integration of existing techniques** (such as CoT or preference fine-tuning). Rather, it represents a novel shift from answer-centric or vision-text alignment to **rationale-centric alignment**. Specifically, SRC differs fundamentally from prior approaches in three key aspects:
>
> 1. **Rationale-Oriented Preference Calibration:** Unlike traditional preference-based fine-tuning methods that solely focus on output correctness or vision-text consistency, ***SRC uniquely prioritizes the internal quality of rationales themselves***. It explicitly calibrates rationale-answer consistency, positioning rationale correctness as a core training objective. We further clarify that ***our Rationale Fine-tuning (Section 3.1) is fundamentally different from standard CoT techniques***. Specifically, CoT explicitly promotes step-by-step reasoning via prompts, whereas our rationale fine-tuning intrinsically induces the model to consistently generate rationale-answer pairs (RAPs) spontaneously, without explicit prompting. Additional detailed discussions distinguishing our approach from CoT are provided in Section 2.
>
> 2. **Iterative Candidate Calibration via Pairwise Scoring:** While existing methods leverage preference fine-tuning broadly, SRC framework innovatively exploits inherent variability among candidate rationale-answer pairs ***through iterative candidate calibration and a tailored pairwise scoring mechanism***. This strategic design effectively discriminates subtle rationale quality differences, accurately capturing relative superiority—even when different answers appear equally at the correctness level.
>
> 3. **Tailored Scoring Model (R-Scorer):** To facilitate efficient, scalable, and effective rationale-answer evaluation, we propose R-Scorer—a lightweight model specifically tailored for pairwise candidate scoring. As demonstrated by experiments and human evaluation, ***R-Scorer substantially outperforms generic LLMs***, even those substantially larger in scale (up to 48×), thereby underscoring the unique advantage of our designed pair-wise scoring strategy.
>
>
> We hope this clarification can addresses the reviewer’s concerns and effectively highlights the originality and contributions of our work.
>
> ---
>
> > **The style and formatting of the main paper.**
>
> We appreciate your feedback on the paper's style and formatting. In the revised version, we will streamline the overall presentation style of our paper. For the citation style, we follow the same format used in previous ICML publications. We sincerely thank the reviewer for your valuable suggestions, which help improve our manuscript.

---

> > ### Comment · Reviewer_CYZ4 · 2025-04-03
> >
> > Thanks for the clarification. The author's responses aligned with my initial understanding of the work; thus, I maintained my original overall recommendation.

---

### Official Review · Reviewer_djqg · 2025-03-14

**Overall Recommendation:** 3

**Summary:**

This paper proposes Self-Ratationale Calibration, a novel framework to align the rationales and answers and LVLMs. SRC shows consistent improvement on both LLaVA-1.5 and LLaVA-Next on several benchmarks.

**Claims And Evidence:**

Yes

**Essential References Not Discussed:**

NaN

**Experimental Designs Or Analyses:**

Yes

**Methods And Evaluation Criteria:**

Yes

**Other Comments Or Suggestions:**

NaN

**Other Strengths And Weaknesses:**

My main concern is about the benchmark results.
1. In Table 1, some results of previous methods are lower than the number from their original paper. For example, RLAIF-V gets 35.4 on MMStar from its original paper, but the authors reported it as 33.7. Similarly, CSR gets 71.1 on LLaVA-Wild, not 64.5. I think this is a serious problem, and the authors should explain the reason for the misalignment in the rebuttal.

2. There are only a few benchmarks reported in the main table. This also hinders a fair judgment about the real performance of the proposed method.

3. A minor point. There are too many details introduced in the method part, a slightly simplified version may be better for readers to capture the main idea of each component.

In a word, I think the proposed method is complex and novel. But the experiment results with the above-mentioned problems failed to prove its effectiveness. I will adjust my final score based on the rebuttal and comments from other reviewers.


## update after rebuttal
The rebuttal partly solved my concerns. I raise my score to weak accept.

**Questions For Authors:**

NaN

**Relation To Broader Scientific Literature:**

NaN

**Theoretical Claims:**

NaN

---

> ### Author Rebuttal · Authors · 2025-03-31
>
> Dear reviewer, due to **space limits** of initial rebuttal, we are unable to elaborate on details or minor points, but we would be glad to clarify any further concerns in the next-round reply.
>
> ---
>
> > **The discrepancies in reported benchmark results.**
>
> We sincerely appreciate your careful examination of our reported benchmark results (Table 1). We understand your concerns about the discrepancies in the reported performance of previous works, particularly the differences in RLAIF-V and CSR results compared to their original papers.
>
> **First**, we would like to clarify **LVLMs are sensitive to deployment environments** (e.g., deployment configurations and inference backends). To ensure fairness and reproducibility, **we consistently evaluated all baseline models using their officially released weights under identical and controlled experimental settings via the VLMEval framework**.
>
> **Second**, regarding the specific cases you highlighted:
>
> For **RLAIF-V**, while the original MMStar score (35.4) is indeed higher than ours (33.7), we noticed **this discrepancy is not unique to our evaluation**. There are other works, such as [1], that report a **much lower value** of 31.8.
>
> For **CSR**, the authors did not release results for LLaVA_Bench (LLaVA-Wild). Additionally, since LLaVA-Wild involves "LLM-as-a-judge" through proprietary GPT-4 API, it is challenging to pinpoint the cause of the observed differences. Notably, **we found significant discrepancies in CSR's original reporting (e.g., SEEDBench) relative to the evaluations of other papers**, such as [2], **whose results closely match ours**:
>
> |     | Name | MMStar | SEEDBench |
> | --- | --- | --- | --- |
> | CSR (official) | LLaVA-1.5-7B | -   |  ***58.6*** |
> | Paper [2] | LLaVA-1.5-7B | 32.2 | 65.6 |
> | Ours | LLaVA-1.5-7B | 32.1 | 64.6 |
> | CSR (official) | LLaVA-1.5-7B + CSR | -   | ***60.3*** |
> | Paper [2] | LLaVA-1.5-7B + CSR | 32.4 | 65.4 |
> | Ours | LLaVA-1.5-7B + CSR | 32.7 | 64.4 |
>
> To further enhance transparency, we will release our evaluation settings and inference results in the future, allowing the community to replicate our results independently. We hope this explanation helps clarify the situation.
>
> [1] A Topic-level Self-Correctional Approach to Mitigate Hallucinations in MLLMs
>
> [2] Self-Correction is More than Refinement: A Learning Framework for Visual and Language Reasoning Tasks
>
> ---
>
> > **More evaluation on various benchmarks.**
>
> Thank you for your valuable feedback. We appreciate your concern regarding the limited number of benchmarks reported in the main table.
>
> We would like to clarify that **MMStar itself integrates multiple comprehensive benchmarks**, including MMBench, SEEDBench, MathVista, and MMMU, all of which address data leakage issues and adhere to the vision-centric QA principle [3]. As such, MMStar can provides a thorough and holistic evaluation of the model's overall capabilities.
>
> In response to your comment, we have conducted **additional evaluations of the following benchmarks**: SEEDBench, AI2D, ScienceQA, and RealworldQA. Please refer to the updated results in the following table:
>
> | Name | SEEDBench | AI2D_TEST | SQA | RealworldQA |
> | --- | --- | --- | --- | --- |
> | LLaVA-1.5-7B | 64.6 | 51.4 | 66.3 | 53.8 |
> | + POVID | 64.1 | 50.7 | 66.1 | 54.2 |
> | + HA-DPO | 63.3 | 50.1 | 65.1 | 53.4 |
> | + SIMA | 64.5 | 47.7 | 66.1 | 52.8 |
> | + SeVA | 63.7 | 49.7 | 64.5 | 54.0 |
> | + RLAIF-V | 64.3 | 51.5 | 63.8 | 50.1 |
> | + CSR | 64.4 | 51.0 | 65.7 | **54.6** |
> | **+ Ours** | **67.3** | **55.4** | **68.1** | 53.9 |
>
> From these extended evaluations, we observe that **our method consistently outperforms prior methods** across most benchmarks, except for RealworldQA (where results are comparable to them). Overall, these additional results support our claim regarding the broad effectiveness and robustness of our approach.
>
> [3] Are we on the right way for evaluating large vision-language models?
>
> ---
>
> > **Providing a slightly simplified version for methodology.**
>
> Thank you for your valuable feedback. We appreciate your suggestion regarding simplifying the method section to enhance readability. In the revised version, we will streamline the descriptions while preserving the key details to ensure clarity for the readers.

---

> > ### Comment · Reviewer_djqg · 2025-04-03
> >
> > The rebuttal partly solved my concerns. I raise my score to weak accept.

---

### Official Review · Reviewer_wbrn · 2025-03-17

**Overall Recommendation:** 4

**Summary:**

The paper introduces Self-Rationale Calibration, a framework designed to enhance the alignment between rationales and answers in VLMS. The motivation stems from the observation that LVLMs can generate correct answers but often fail to provide factually grounded rationales, leading to inconsistent reasoning. Generally speaking, SRC calibrates LVLMs by iteratively aligning rationales with answers, improving logical consistency. Additionally, the authors introduce R-Scorer, a lightweight LLM-based evaluator that scores responses based on rationale quality and factual consistency. Finally, experimental results across multiple VQA benchmarks indicate that SRC outperforms existing alignment methods, improving both perception and logical reasoning capabilities.

**Claims And Evidence:**

They are supported by qualitative examples and quantitative improvements in fine-grained perception and logical reasoning.

**Essential References Not Discussed:**

n/a

**Experimental Designs Or Analyses:**

The experiments compare SRC with state-of-the-art LVLM post-training strategies, including DPO-based methods (e.g., RLAIF-V, CSR, SeVA).
Results indicate SRC provides significant improvements in:
- Logical reasoning: +8.4% increase (from 39.6 → 43.6 on MMStar)
- Fine-grained perception: +8% increase (25.2 → 33.2)
- Math reasoning: +9.2% improvement
Ablation studies confirm the importance of rationale fine-tuning, scoring, and iterative alignment.

One concern is that Iterative fine-tuning with preference alignment is expensive, requiring multiple rounds of scoring and calibration.

**Methods And Evaluation Criteria:**

Yes, the proposed methods and/or evaluation criteria make sense for the problem.

**Other Comments Or Suggestions:**

There are some other papers about rationalization, would it be possible to discuss them in related work?

[1] Decoupled Rationalization with Asymmetric Learning Rates: A Flexible Lipschitz Restraint [2] Is the MMI Criterion Necessary for Interpretability? Degenerating Non-causal Features to Plain Noise for Self-Rationalization [3] Breaking Free from MMI: A New Frontier in Rationalization by Probing Input Utilization [4] MGR: Multi-generator Based Rationalization.

**Other Strengths And Weaknesses:**

S:
- SRC explicitly aligns rationales with answers via preference fine-tuning, a novel improvement over DPO.
- Provides an alternative to costly GPT-4o-based evaluations.

W:
- SRC involves multiple iterations of fine-tuning and preference scoring. It would be better to discuss the efficiency in detail.

**Questions For Authors:**

n/a

**Relation To Broader Scientific Literature:**

Related to multimodal grounding approaches.

**Theoretical Claims:**

The paper assumes that rationale fine-tuning inherently improves answer quality, but it does not explore failure cases or potential biases introduced by the fine-tuning process.

---

> ### Author Rebuttal · Authors · 2025-03-31
>
> Dear reviewer, due to **space limits** of initial rebuttal, we are unable to elaborate on details or minor points, but we would be glad to clarify any further concerns in the next-round reply.
>
> ---
>
> > **Training efficiency of SRC.**
>
> We sincerely appreciate the reviewer's feedback considering the training efficiency of our SRC framework.
>
> **First**, we would like to clarify that direct efficiency comparisons across recent post-training methods are challenging, **due to the diversity in methodology and resource requirements**. For instance, methods such as RLAIF-V requires deployment of multiple open-sourced LVLMs; POVID and HA-DPO depend heavily on proprietary models (e.g., GPT-4V) for generating preference data; and CSR, the most comparable method to ours, also employs an iterative post-training strategy.
>
> **Second**, regarding the efficiency of our framework, although SRC may not match the efficiency of methods using proprietary LVLMs (API-driven) and single-round preference fine-tuning, e.g., POVID or HA-DPO, **SRC consistently achieves substantial improvements** in perception, reasoning, and generalization across various benchmarks. Moreover, the iterative nature of SRC represents an deliberate design choice aimed at progressively enhancing alignment between rationales and answers. Importantly, as demonstrated in Figure 7, **even a single iteration of SRC yields significant performance improvements** across benchmarks. This highlights that substantial gains can be achieved early in the process, offering a favorable option/trade-off between computational cost and performance enhancement.
>
> **Thrid**, to proactively address and mitigate the efficiency concern associated with the iterative process, we **have incorporated several key optimizations** into the SRC framework:
>
> 1. **Optimized Candidate Generation**: SRC employs ***sentence-level beam search with constrained search-tree widths*** to generate rationale-answer pair (RAP) candidates. This approach significantly reduces computational overhead compared to exhaustive beam search. Please refer to Appendix A.2 ("Candidate Generation") for further details.
>
> 2. **Lightweight Pairwise Scoring Model**: Instead of utilizing generic LLMs, which can be substantially larger (up to approximately 48×), SRC introduces R-Scorer, ***a specialized lightweight scoring model*** tailor for evaluating rationale quality and factual consistency in a pairwise socring manner. Our experiments and human evaluations (Figure 6) demonstrate R-Scorer’s superior balance of efficiency and effectiveness in the scoring process.
>
> 3. **Efficient Engineering Implementation:** SRC incorporates the vLLM library with the prefix caching technique for ***accelerating inference during scoring***, further enhancing computational efficiency.
>
>
> In summary, SRC's computation cost is comparable to similar iterative methods, and significant performance benefits can be observed even after a single iteration. The combination of improved candidate generation, a lightweight specialized scoring model, and optimized inference implementation ensures that SRC provides a practical and effective approach for enhancing LVLMs' alignment between rationales and answers.
>
> ---
>
> > **More discussion of related works.**
>
> We thank the reviewer for suggesting the inclusion and discussion of recent rationale-focused works.
>
> For [1] and [4], they focus primarily on addressing internal degeneration problems within the rationale generation process itself, while [2] and [3] explore rationale extraction based on the maximum mutual information (MMI) criterion, aiming specifically at mitigating spurious feature reliance ([2]) and probing model input utilization ([3]).
>
> Here, we consider [2] and [3] to be more closely aligned with the context of SRC. [2] and [3] target input-level rationalization (selecting input subsets as rationales), whereas the proposed Self-Rationale Calibration (SRC) framework operates distinctly **at the output level**, calibrating **the alignment between generated rationales and answers** within LVLMs.
>
> In the revised version, **we will incorporate references [2] and [3] in our paper**, highlighting the methodological differences from SRC and discussing their broader implications for rationale-aware modeling. We greatly appreciate this valuable suggestion and believe such clarifications will further strengthen the presentation of our contributions.
>
>
> [1] Decoupled Rationalization with Asymmetric Learning Rates: A Flexible Lipschitz Restraint
>
> [2] Is the MMI Criterion Necessary for Interpretability? Degenerating Non-causal Features to Plain Noise for Self-Rationalization
>
> [3] Breaking Free from MMI: A New Frontier in Rationalization by Probing Input Utilization
>
> [4] MGR: Multi-generator Based Rationalization

---

> > ### Comment · Reviewer_wbrn · 2025-04-02
> >
> > Thank you for the reponse and I have raised the score accordingly.

---

### Official Review · Reviewer_Knwn · 2025-03-21

**Overall Recommendation:** 3

**Summary:**

This paper attempts to address the misalignment between the final answers and the perceptual reasoning, i.e., rationales, from LVLMs' outputs. With a prior fine-tuning for the model to generate rationales, the authors propose a pairwise scoring strategy considering model confidence and LLM-driven assessment, i.e., R-scorer, to identify superior preference pairs for fine-tuning the quality of the rationale-grounded responses. Extensive experiments have validated that the alignment is able to allow overall perceptual improvement across various task domains.

## update after rebuttal
Considering other reviewers' comments, I agree that there exists some concerns regarding theoretical novelty. I decide to keep my original recommendation.

**Claims And Evidence:**

The main claim that rationale-response alignment improves the model’s perceptual ability, is clearly and comprehensively validated through experiments.

**Essential References Not Discussed:**

Key references are well-discussed.

**Experimental Designs Or Analyses:**

It seems the Table 1’s Math and S&T results for LLaVA-1.5 uses different $\alpha$ values to achieve their best shown in Fig. 9. Such inconsistency would degrade the credibility of the work.

**Methods And Evaluation Criteria:**

Yes, the method is novel and appear to be well-aligned with the problem. The evaluated benchmark includes various ability tests for LVLMs.

**Other Comments Or Suggestions:**

1. The two types of winning scores have the same notation in Eq.1 and Eq. 2.

2. Typos:
(a) Line 131 right column: “provide a *rationales* before providing *a* answer”.
(b) Line 270-271 right column: “then *sent* them”.

**Other Strengths And Weaknesses:**

Strengths:

1. The problem identified exists in most LVLMs and is indeed a crucial challenge for visual understanding. The proposed method has verified that aligning language reasoning & results can also improve visual perceptual ability of LVLMs, demonstrating great novelty.

2. The light-weight scoring model is effective and efficient in identifying superior preference pairs.

3. The experimental results demonstrate comprehensive improvements over various domains, verifying the advanced visual perception.


Weaknesses:

1. The beam search results may have similar confidence scores as they are top log-probability results. With such similarity, I doubt the usability of the confidence scores. In the appendix, the overall distribution visualizations of the preferred & non-preferred responses tell little about whether confidence scores help to distinguish candidates in an individual response group.

2. The scale of x-axis in Fig. 9(a) is weird: [0.0, 0.2, 0.4, 0.6, 0.8, **0.9**, 1.0]. The authors could re-organize the grid intervals to accommodate the 0.9 results.

3. Currently, there are only limited results on LLaVA-Next-8B with no comparison to other methods. Results on architectures other than LLaVA are also limited.

4. Also see the above *Experimental Designs Or Analyses* for my concern about the consistency issues.

**Questions For Authors:**

1. It’s unclear what the calibration process uses as the “remaining data” stated in Sec. 4.1 “Evaluation”.

2. Why does R-scorer perform better against larger LLMs while having much less parameters? The authors are suggested to include some analysis in Sec. 4.3 “Different Scoring Models”.

**Relation To Broader Scientific Literature:**

The work is closely related to LLM-based judge and preference optimization of LVLMs, which are referenced in the main paper. Meanwhile, the data used in this work involves reasoning ideas such as chain-of-thought reasoning.

**Theoretical Claims:**

The paper does not involve any theoretical results.

---

> ### Author Rebuttal · Authors · 2025-03-30
>
> Dear reviewer, due to **space limits** of initial rebuttal, we are unable to elaborate on details or minor points, but we would be glad to clarify any further concerns in the next-round reply.
>
> ---
>
> > **The usability of the confidence scores in Confidence-weighted Winning Score.**
>
> While we acknowledge that sentence-level beam search may yield candidates with closely ranked log-probability scores, we would like to clarify the following points:
>
> 1. **The confidence score is a complementary role in the candidate selection process.** The core of SRC’s candidate selection lies in the ***pairwise scoring*** judged by the R-Scorer. The introduction of confidence score can be considered as a ***post-processing*** of winning scores against edge cases (e.g., ties between candidates), as discussed in Section 3.3. The results in Figure 9 that the moderate introduction of confidence ($1-\alpha$) benefits SRC performance, also support the effectiveness of this design choice.
> 2. **SRC adopts a relative confidence advantage in the candidate selection process.** Even when beam candidates have similar confidence scores, their ***relative ranking*** remains informative. Specifically, we transform log-probabilities into rank-based scores (Equation 4) and apply a rank-based weighting scheme (Equation 5). This allows the framework to capture their ordinal confidence advantage, allowing for more nuanced processing of winning scores even when the confidence scores are similar.
>
> We hope this addresses your concerns regarding the usability of confidence scores in candidate selection.
>
> ---
>
> > **Results on LLaVA-Next-8B and more experiments on other LVLM architectures.**
>
> We sincerely appreciate the reviewer’s valuable suggestion.
>
> For existing methods (e.g., CSR, RLAIF-V, and SeVA), they are **completely built upon the LLaVA-1.5 codebase**. As these methods **have not been adapted to more advanced architectures** like LLaVA-Next, our primary experiments focused on LLaVA-1.5 for fair comparison.
>
> Here, we have also conducted **preliminary experiments with Qwen-VL [1]**. Though limited by the rebuttal time for optimal tuning, the initial results (shown below) support SRC's generalizability across different architectural paradigms:
>
> | Method | Overall | CP | FP | IR | LR  |
> | - | - | - | - | - | - |
> | Qwen-VL  | 35.7    | 60.8 | 30.0 | 49.2 | 27.6 |
> | **+ SRC (ours)** | **38.8** | **64.4** | **35.6** | **50.0** | **29.2** |
>
> [1] Qwen-VL: A Versatile Vision-Language Model for Understanding, Localization, Text Reading, and Beyond
>
> ---
>
> > **The inconsistency of the main results and Figure 9.**
>
> We sincerely appreciate the reviewer’s kind attention of our results. We would like to clarify that the discrepancy between Table 1 and Figure 9 **arises from their different experimental settings**:
>
> * Table 1 (the main results) reports **final post-training performance** after full SRC post-training, where LVLMs are undergone multiple iterations as detailed in Section 3.4.
> * Figure 9 shows ablation results of **intermediate iteration** since considering the efficiency of the ablation studies.
>
> Below we provide **the experiment results ($\alpha=\{0.8, 0.9\}$, and full training)** for reference:
>
> | Training | Math | S&T |
> | - | - | - |
> | 0.8 (1 iter, Figure 9) | 30.0 |  **29.6**|
> | 0.9 (1 iter, Figure 9) |  **34.0** | 24.8  |
> | Full SRC, Table 1 |  33.2 | 28.8 |
> (overall performance of full SRC is the best)
>
> We will clearly state the experiment setting difference in the revised version and apologize for any confusion caused by this omission.
>
> ---
>
> > **General LLMs and R-Scorer during the pair-wise scoring process.**
>
> We appreciate the reviewer’s comment on R-Scorer’s strong performance despite its smaller size. While large generic LLMs (e.g., Qwen-72B, LLaMA-3-70B) are trained for broad tasks (e.g., dialogue, math, QA), **R-Scorer is specifically optimized for pairwise scoring in SRC**. Its focused training enables more effective evaluation of rationale quality and factual consistency. This is the reason why its lightweight scale can outperform large generic LLMs in SRC, as supported by our experiments.
>
> As shown in Table 5 (Appendix), scaling R-Scorer (1.5B → 7B) indeed improves post-training performance. However, considering **the trade-off between efficiency and marginal gains**, we chose the 1.5B model for formal experiments.
>
> ---
>
> >  **The clarification of the “remaining data”.**
>
> We acknowledge that the term "remaining data" was ambiguous and will restructure Section 4.1 to explicitly delineate data usage across all stages. Below is a detailed clarification:
>
> **Data Pipeline:**
> * Our initial data pool comprises 57K samples collected from open-source datasets.
> *  Through rationale augmentation and filtering (Section 3.1), we curated the final 43K samples for SRC.
>
> **Data Allocation:**
> * Rationale Fine-tuning: ~20K samples (Section 3.1).
> * Preference Fine-tuning: 12K samples for calibration (Section 3.4).
> * Evaluation: The remaining portion.

---

> > ### Comment · Reviewer_Knwn · 2025-04-03
> >
> > Thank you for the clarifications. I raise my score to accept.

---

### Decision · Program_Chairs · 2025-05-01

**Decision:**

Accept (poster)

**Comment:**

This paper introduces Self-Rationale Calibration (SRC), a framework designed to improve the alignment between rationales (reasoning steps) and final answers in Large Vision-Language Models (LVLMs). The key contributions include: (1) a rationale fine-tuning stage to encourage coherent reasoning, (2) a lightweight R-Scorer for pairwise preference scoring, and (3) an iterative alignment process combining confidence-weighted scoring and preference optimization. Empirical results demonstrate improvements across multiple benchmarks (e.g., MMStar, SEEDBench, ScienceQA), suggesting that better rationale-answer alignment enhances both perceptual and reasoning capabilities in LVLMs.

All the reviewers agreed that the empirical performance of this paper is solid. However, there exist several key concerns raised by the reviewers. From the perspective of novelty and theoretical contribution, the contribution could be limited since the core methodology (rationale fine-tuning + preference scoring) is regarded as an integration of existing techniques. Additionally, there exist discrepancies in baseline performance (e.g., RLAIF-V, CSR), and limited comparisons to other architectures raised questions about generalizability.

During the rebuttal phase, some of the key concerns have been addressed. The authors attributed baseline discrepancies to evaluation settings and provided additional results (e.g., Qwen-VL) to support generalizability. However, some concerns persisted, such as theoretical novelty. Most of the reviewers agreed that this paper presents a practically useful framework with strong empirical evidence, while the conceptual novelty is limited, which makes this paper close to borderline. Given these considerations, AC recommends weak acceptance. The authors are encouraged to improve the paper by considering the reviewers’ comments carefully.